# FedEMoE: Improving Personalization on Heterogeneous Federated Learning via Elastic Mixture of Experts Architecture

**Haizhou Du** [1]  **Lixin Huang** [1]  **Zonghan Wu** [2]  **Huan Huo** [3]

## Abstract

Heterogeneous federated learning (HtFL) has emerged as a promising approach to address heterogeneity in local computational resources and data distribution. However, existing methods cause performance degradation of model personalization because personalized and generalized knowledge are either intertwined or dominated by one of them. To address this issue, we propose a novel **E**lastic **M**ixture **o**f **E**xperts (EMoE) architecture on HtFL, namely FedEMoE, decoupling personalization from generalization. Specially, FedEMoE employs a multi-scale feature extraction mechanism via personalized experts to enrich personalized knowledge. Furthermore, we design an elastic shared expert to break the transferred knowledge bottleneck across heterogeneous client models. The elastic shared expert can adaptively expand or shrink according to the status of each expert by the weight spectrum analysis, respectively. Extensive experiments across statistical and model heterogeneity settings demonstrate that FedEMoE significantly outperforms state-of-the-art methods on the accuracy of each heterogeneous model over diverse datasets.

## 1. Introduction

Heterogeneous federated learning (HtFL) has emerged as a promising paradigm for enabling multiple clients to collaboratively enhance their local heterogeneous model where data is naturally distributed and highly heterogeneous as common in the real world (Li et al., 2020; Kairouz et al., 2021). Model heterogeneity in HtFL, arising from clients employing customized architectures for individual requirements, fundamentally prevents the direct knowledge sharing across clients. Simultaneously, statistical heterogeneity leads to client model drift that slows convergence and degrades final performance. When the above bottlenecks are overcome, HtFL can break down data silos and boost knowledge sharing in collaborative learning.

Existing HtFL methods address these constraints by three ways. Knowledge distillation-based methods (Lin et al., 2020; Wu et al., 2022) share softened predictions to align client models, improving collaborative learning without exchanging raw model parameters. Prototype-based methods (Zhang et al., 2024; Zhou et al., 2025) compress local representations into class-wise vectors, enabling efficient knowledge sharing across clients. Model-based methods (Niu & Deng, 2022; Yi et al., 2025) share and jointly update common model components, thereby supporting both global adaptation.

However, existing methods force personalization and generalization within the same set of parameters, resulting in a decline in local personalization. We can obviously summarize two *challenges* as follows: *i) How to decouple models' generalization and personalization is the key important challenge.* This allows each of them to have a parameter space, which avoids the zero-sum game dilemma (Chen et al., 2024). *ii) How to construct a knowledge-rich shared repository for client-cross personalization is an important challenge.* The existing knowledge transfer mechanisms—whether logits, prototypes, or averaged parameters— become a compromise that cannot satisfy the diverse needs for personalized enhancement across clients (Zhu et al., 2024; Meng et al., 2025).

This motivates our core idea: decoupling personalization from generalization at the architectural level and shared knowledge level. Specifically, at the architectural level, we decompose the local model into a personalized component and generalized one via a Mixture-of-Experts (MoE) framework. This physical separation of parameters terminates the zero-sum competition by providing dedicated subspaces. At the shared knowledge level, we design a dynamic and rich knowledge repository to improve client-cross personalization. It integrates personalized knowledge from clients into a generalized yet pattern-diverse knowledge repository.

---

[1]Shanghai University of Electric Power, Shanghai, China [2]East China Normal University, Shanghai, China [3]University of Technology Sydney, Sydney, Australia. Correspondence to: Haizhou Du <duhaizhou@shiep.edu.cn>.

*Proceedings of the $43^{rd}$ International Conference on Machine Learning*, Seoul, South Korea. PMLR 306, 2026. Copyright 2026 by the author(s).

Building on this insight, our **main contributions** are:

- To the best of our knowledge, we first design a novel elastic MoE architecture for HtFL to decouple personalization from generalization, namely FedEMoE, ensuring local models never compromise their specialization for global consensus.

- We introduce a multi-scale feature extraction and knowledge exchange mechanism based on a MoE architecture at client-side. This mechanism maximizes the feature captured from local unique data and retained the ability to exchange knowledge with other clients, resulting in the improvement of personalization.

- We design an elastic MoE architecture where the shared expert acts as a dynamic repository of collective intelligence at server-side. Its structure can adaptively strengthen its representation capacity by preserving and integrating specialized knowledge rather than averaging it away.

- We evaluate FedEMoE in settings with model and statistical heterogeneity. Our extensive experiments and ablation studies demonstrate that FedEMoE is superior to state-of-the-art methods, improving task accuracy by up to 48.05%.

**Conflict of Interest Disclosure:** The authors declare that they have no financial conflicts of interest related to this work. This research was conducted purely as an academic work, without any involvement of industry or commercial entities.

## 2. Related Work

We classify the existing methods of HtFL into three categories: *knowledge distillation-based, prototype-based* and *model-based methods*.

**Knowledge Distillation-Based FL Methods.** Knowledge distillation(KD)-based FL methods use model outputs (logits), so they are suitable for model-heterogeneous scenarios. Some methods, such as FeAUX (Sattler et al., 2021), FedDF (Lin et al., 2020), and DS-FL (Itahara et al., 2021), enable models to learn from others by identifying differences in model outputs on a global dataset. To overcome the limitations of public datasets, some approaches have been proposed: FedGen (Zhu et al., 2021) uses a global generator based on clients' outputs to generate data, and FedKD (Wu et al., 2022), FML (Shen et al., 2020) advance KD-based methods in a data-free manner, where they share a small model as knowledge, rather than relying on a global dataset. However, these methods incur additional computational overhead or require a high-quality public dataset,

thereby constraining their applicability, and smooth out specific decision boundaries.

**Prototype-Based FL Methods.** Prototype-based FL approaches utilize local data feature representations. For example, FedPAC (Xu et al., 2023), FedGKT (He et al., 2020), and FedProto (Tan et al., 2022) upload local class-specific representations or prototypes to the server to gain global knowledge. FedTGP (Zhang et al., 2024) introduces a regularization approach to increase the spacing between classes, enhancing prototype distinctiveness. FedSA (Zhou et al., 2025) employs semantic anchors to align global prototypes within a unified feature space. However, these methods incur substantial computational overhead, discard fine-grained local structure, and produce increasingly indistinct global prototypes as client numbers grow, significantly impairing training efficiency.

**Model-Based FL Methods.** The main concept of model-based FL approaches is to divide client models into two parts: one for global aggregation and sharing, and another one retained locally for model personalization. For instance, methods like FedGC (Niu & Deng, 2022), FedROD (Chen & Chao, 2022), FedRep (Collins et al., 2021), FedBABU (Oh et al., 2022), and FedAlt (Pillutla et al., 2022) split the model into homogeneous feature extractors and heterogeneous classifiers, enabling data mapping to a consensus feature space and preserving local personalization through classifiers. In contrast, methods such as LG-FedAvg (Liang et al., 2020), and Fedclassavg (Jang et al., 2022) use heterogeneous feature extractors and homogeneous classifiers, achieving global knowledge acquisition by sharing classifiers for unified classification standards. However, these methods force local models to compromise their personalization for the sake of global agreement.

## 3. Methodology

### 3.1. Problem statement

We consider a federated learning system comprising a central server and K heterogeneous clients. Each client $i$ possesses a private, potentially non-IID dataset $D_i$ and maintains a personalized model $M_i$ with a potentially unique architecture. The goal is to collaboratively improve all local models by facilitating knowledge transfer through the server, while strictly bounding communication overhead. Formally, we aim to solve:

$$\min_{\{M_i\}} \frac{1}{K} \sum_{i=1}^{K} \mathcal{L}(M_i, D_i) \ \text{ subject to } \ \frac{1}{K} \sum_{i=1}^{K} \mathcal{C}(M_i) < c,$$

(1)

where $\mathcal{L}(\cdot, \cdot)$ is the loss function, $\mathcal{C}(\cdot)$ measures the communication cost of transmitting model $M_i$ and $c$ is the communication budget. The local update for client $i$ follows the

standard gradient descent step:

$$M_i^{t+1} = M_i^t - \eta \nabla_{M_i} \mathcal{L}(M_i^t, \mathcal{D}_i), \qquad (2)$$

where $\eta$ is the local learning rate and $\mathcal{D}_i \subseteq D_i$ is the training batch.

## 3.2. The Overview of FedEMoE

As illustrated in Figure 1, each client model $M_i$ in FedEMoE consists of two key components and a knowledge exchange mechanism: i) A personalized expert group (PEG) contains a set of $N_i$ heterogeneous expert networks. These experts are not shared and are specialized to capture multi-scale, client-specific patterns from local data. ii) A shared expert (SE) acts as a shared knowledge hub. Its capacity can adaptively expand or shrink based on collective training. PEG and SE are two parallel modules that decouple personalization and generalization at the model architecture level. iii) A distillation-based knowledge exchange mechanism between PEG and SE improve local knowledge transfer.

On the server-side, we aggregate FEs and SEs from participating clients in each round, respectively. Then we propose an elastic MoE architecture to keep the SE always in a healthy status and reduce unnecessary computation and communication costs in Section 3.4.

## 3.3. Multi-Scale Feature Extraction Expert

We specifically design Personalized Expert Group (PEG) to enhance local personalization. Moreover, the design of the PEG is motivated by the heterogeneity of data distributions across clients and the difficulty for a single fixed-scale feature to adapt to samples from different tasks. Inspired by this, we employs multiple parallel feature extraction networks to capture client-specific patterns at multiple scales. Specifically, we design PEG as an ensemble of $N_i$ architecturally diverse expert networks $\{P_n\}_{n=1}^{N_i}$, each specialized in extracting features at distinct semantic scales.

Formally, given an input feature representation $h \in \mathbb{R}^d$ produced by the shared feature extractor, each expert $P_n$ independently processes $h$ to generate a scale-specific feature vector:

$$\mathbf{z}_n = P_n(\mathbf{h}) \in \mathbb{R}^{d_n}, \quad n = 1, \ldots, N_i, \qquad (3)$$

where $d_n$ denotes the output dimension of expert $n$. The multi-scale feature is then constructed through feature concatenation across all experts:

$$\mathbf{z} = [\mathbf{z}_1 \oplus \mathbf{z}_2 \oplus \cdots \oplus \mathbf{z}_{N_i}] \in \mathbb{R}^{\sum_{n=1}^{N_i} d_n}, \qquad (4)$$

where $\oplus$ denotes the concatenation operator. This concatenated feature vector $\mathbf{z}$ preserves the complete multi-scale information without early fusion, allowing subsequent personalized headers to learn optimal combinations of scale-specific features.

## 3.4. Elastic MoE Architecture

Our core design differs from traditional MoE architectures, where we expect each expert to specialize in specific data distributions rather than aiming for traditional load balancing. We think shared module is generalized on the whole yet personalized in its fine-grained parts, rather than averagely aggregated personalization module. Given that MoE's sparsity satisfies our requirements for the shared module, we propose the elastic MoE (EMoE) architecture based on the MoE architecture. We expect each expert in EMoE architecture to specialize in specific data distributions rather than aiming for traditional MoE load balancing . We apply this architecture to shared expert and let shared expert become a personalized knowledge shared repository. However, the fixed number of experts means fixed partial personalization capability. We design a mechanism to dynamically adjust the number of experts. This EMoE architecture works through the following three components.

### 3.4.1. SPARSE MODEL AGGREGATION

We adopt a sparse aggregation strategy for knowledge sharing. Specifically, during each communication round, participating clients train their local model and a subset of sub-experts in SE are activated due to the gating mechanism of EMoE. Only the activated sub-experts are uploaded to the server. On the server side, for each sub-expert $j$, aggregation is performed exclusively over the subset of clients that activated it. Formally, $\mathcal{K}_j \in \{1, 2, \ldots, K\}$ denotes the set of clients that activated sub-expert $j$, the updated global sub-expert $SE_j^{(t+1)}$ is computed as follows:

$$SE_j^{(t+1)} = \frac{1}{\sum_{i \in \kappa_j} |\mathcal{D}_i|} \sum_{i \in \kappa_j} |\mathcal{D}_i| \cdot SE_{j,i}^{(t)}. \qquad (5)$$

This sparse aggregation mitigates the over-averaging of each sub-expert. Specifically, aggregating with inactive sub-experts $\{SE_j | j \notin K_j\}$ would bias the updated sub-expert $SE_j^{(t+1)}$ toward its previous state $SE_j^{(t)}$, thereby reducing the sharing of newly acquired knowledge. Each client favors a subset of sub-experts due to routing mechanism, creating local personalization. However, from a global perspective, the SE remains generalizable. Moreover, since each client only uploads its activated sub-experts rather than the entire SE, sparse aggregation reduces communication overhead.

### 3.4.2. EXPERT STATUS DIAGNOSIS

We introduce the weight spectrum analysis technique (Yunis et al., 2025; Martin & Mahoney, 2021) as a diagnostic tool to monitor the status of sub-experts and decide when to alter model's structure. Specifically, we perform singular value decomposition (SVD) on the weight matrix $\mathbf{W}^{(l)} \in \mathbb{R}^{m \times n}$

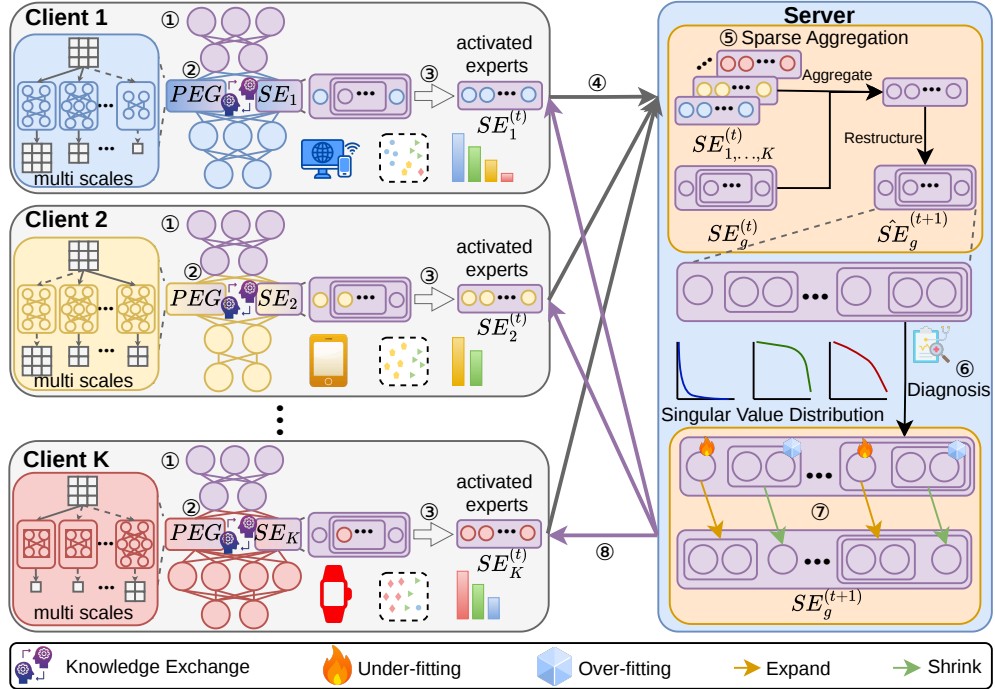

*Figure 1.* The architecture of FedEMoE. ① Local update. ② Knowledge exchange between personalized expert group (PEG) and shared expert (SE). ③ The clients filter the activated experts in its own SE. ④ Participating clients upload feature extractors (FE) and SE. ⑤ The server executes weighted aggregation of FE and sparse aggregation of SE. ⑥ Server diagnoses the sub-experts of the aggregated SE based on weight spectrum analysis. ⑦ The sub-experts of the aggregated SE respectively execute the expansion or shrinkage according to the diagnosis results. ⑧ The server sends global FE and global SE to clients, and clients use them to replace their local ones.

of each layer $l$ and sort its singular values in descending order: $\sigma_1^{(l)} \geq \sigma_2^{(l)} \geq \cdots \geq \sigma_r^{(l)} > 0$. We judge the status of experts through two key indicators: the effective rank $p_\theta^{(l)}$ and the tail energy $T_\alpha^{(l)}$. The effective rank $p_\theta^{(l)}$ is defined as the smallest fraction of full rank required to retain an energy ratio $\theta$, where

$$p_\theta^{(l)} = \min \left\{ k \mid \frac{\sum_{i=1}^k (\sigma_i^{(l)})^2}{\sum_{i=1}^r (\sigma_i^{(l)})^2} \geq \theta \right\}. \qquad (6)$$

The tail energy $T_\alpha^{(l)}$ measures the residual energy after discarding the top $\alpha$ proportion of components, where

$$T_\alpha^{(l)} = \frac{\sum_{i=\alpha+1}^r (\sigma_i^{(l)})^2}{\sum_{i=1}^r (\sigma_i^{(l)})^2}. \qquad (7)$$

As shown in Figure 2, we classify experts into three status based on the shape of the spectrum. *Under-fitting* (Figure 2a), *Normal* (Figure 2b), *Over-fitting* (Figure 2c). Specifically, a rapidly decaying spectrum indicates stable capacity with negligible drift, whereas a heavy-tailed spectrum signals redundancy and overfitting. Unlike validation-loss or gradient-norm triggers, weight-spectrum analysis requires no extra data or auxiliary information and has a provable bound (Appendix D.7.2).

However, when clients activate the same subset of sub-experts, the SE may cause redundant computations due to excessive expansion, without obvious gain in accuracy. In order to limit this expansion, we have adopted gradually stricter expansion conditions. Specifically, we treat the effective rank as a time series $\mathcal{S} = \{ S_t \mid t \in T \}$, where $T$ is the total number of communication rounds. We employ an Exponential Moving Average (EMA) mechanism to gradually adjust the expansion threshold, thereby adaptively suppressing expansions. At round $t$, we allow expansion only if $p_\theta^{(l)}(t) > v_t$, thus blocking further growth once the effective rank stops increasing. The recursive formula for the dynamic threshold $v_t$ is as follows:

$$v_t = \begin{cases} v_0, & t = 1 \\ \gamma S_t + (1-\gamma)v_{t-1}, & t \geq 2 \end{cases}, \qquad (8)$$

where $v_0$ is the initial value as a hyperparameter and $\gamma$ is the smoothing factor. By gradually increasing the expansion threshold, we can limit the disorderly expansion of some sub-experts.

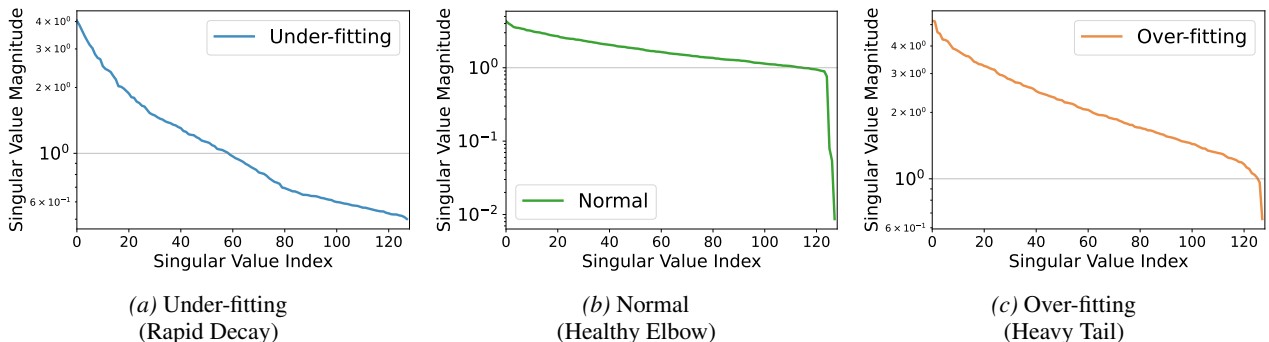

*(a)* Under-fitting
(Rapid Decay)

*(b)* Normal
(Healthy Elbow)

*(c)* Over-fitting
(Heavy Tail)

*Figure 2.* Weight spectrum analysis for expert status classification under CIFAR-100. The singular value decay patterns reveal expert training status: (a) Under-fitting shows rapid decay with low effective rank, (b) Normal exhibits a healthy elbow pattern with balanced capacity, (c) Over-fitting displays heavy tails indicating parameter redundancy. This analysis guides our elastic expansion and contraction decisions.

### 3.4.3. EXPERT EVOLUTION

We design a process of experts evolution to satisfy the richer representation of each SE, where each expert exists in two statuses: an expert $(e)$ represents a single neural network component with capacity (the amount of data it can adapt to), while a MoE-based expert $(e^*)$ represents a composite structure containing $N$ sub-experts $(e)$ with internal routing.

**Shrinkage** $(e^* \rightarrow e)$ is invoked when the $e^*$ is flagged as over-fitting (heavy-tailed spectrum). The $e^*$ is compressed to an expert $e$ via the weighted aggregation, the weight is the global activation rate of each sub-expert in $e*$. This shrinkage removes redundant parameters, cuts memory and communication costs, while retaining the knowledge of the $e^*$. The Algorithm 1 shows the shrinkage process.

---

**Algorithm 1** ShrinkMoE

---

**Input:** Number of experts $N$, Activation proportion vector $B_g = (b_{g,1}, b_{g,2}, \ldots, b_{g,N})^\top$, MoE-based Expert $e^* = (e_1, e_2, \ldots, e_N)^\top$.
**Output:** Shrunk expert $e$
1: $e \leftarrow \mathbf{0}$
2: **for** $r = 1$ **to** $N$ **do**
3:    $e \leftarrow e + b_{g,r} \cdot e_r$
4: **end for**

---

**Expansion** $(e \rightarrow e^*)$ is triggered when $e$ is diagnosed as under-fitting (low effective rank). The original $e$ becomes $e_j \in e^*$, and the remaining experts are randomly initialized. When the expert $e$ in $e*$ meets the expansion conditions, it can also be expanded. This expansion aims to improve the capacity that the under-fitting expert lacks, preserving existing knowledge while adding expressive power, thus improving local accuracy without extra validation data. The specific expansion process can be found in the Algorithm 2. Discussions on the expansion strategy for initializing new experts can be found in the Appendix D.1.

---

**Algorithm 2** ExpandExpert

---

**Input:** Expanding expert $e$, the router $R$ of $e$'s parent-MoE, the index $p$ of $e$ in parent-MoE, number of experts $N$
**Output:** MoE-based Expert $e^* = (R, \mathcal{E})$, where $\mathcal{E} = (e_1, \ldots, e_N)$
1: $\mathcal{E} \leftarrow InitalizeList(N)$
2: **for** $r = 1$ **to** $N$ **do**
3:    **if** $r = p$ **then**
4:        $e_r \leftarrow E$
5:    **else**
6:        $e_r \leftarrow InitializeExpert$
7:    **end if**
8:    $\mathcal{E}[r] \leftarrow e_r$
9: **end for**
10: $e^* \leftarrow (R, \mathcal{E})$

---

**Algorithm 3** FedEMoE

---

**Input:** Total number of clients $K$, Client participation fraction $J$, Initial client models $\{W_i^0\}_{i=1}^K$, Initial global model $W_g$, Communication rounds $T$
1: **for** $t = 1$ **to** $T$ **do**
2:    Select a subset of clients $K_t \subseteq (1, \ldots, K)$, $|K_t| = max(1, JK)$
3:    **for** $i \in K_t$ **do**
4:        $f_i^t, s_i^t \leftarrow$ ClientUpdate($f_g^{t-1}, s_g^{t-1}, M_i^{t-1}, \mathcal{D}_i$)
5:    **end for**
6:    $f_i^t, s_i^t, B_g \leftarrow$ Weighted Aggregation($f_i^t|_{i=1}^{K_t}, s_i^t|_{i=1}^{K_t}, |\mathcal{D}_i|_{i=1}^{K_t}, B_i^t|_{i=1}^{K_t}$)
7:    Every $\mathcal{T}$ do:
8:    Parallel processing of ExpandExpert($e$, $p$, $N$) and ShrinkMoE($N$, $B_g$, $E$)
9: **end for** $f_i^t, s_i^t$

---

Unlike width scaling(Appendix D.7.1), our approach preserves the existing feature space (Theorem 3.1). The convergence analysis (Theorem 3.2) of experts' evolution is presented in Appendix D.5. Through the process, the number of sub-experts can be dynamically changed, and the hierarchical structure of sub-experts enables each sub-expert to serve a small and stable subset of clients.

**Theorem 3.1** (Feature Preservation of Expert Evolution). *Let $\mathcal{F}_j$ be the feature mapping of expert $e_j$, and $\mathcal{H} = span\{\mathcal{F}_j\}$ the feature space.*

*Expansion: Adding expert $e_{N+1}$ satisfies:*

$$\mathcal{H}_{new} = \mathcal{H} \oplus span\{\mathcal{F}_{N+1}\}, \quad \mathcal{F}_j(x) \text{ unchanged } \forall j \leq N. \tag{9}$$

*Shrinkage: Merging experts via $\hat{W} = \sum a_j W_j$ yields:*

$$\|\hat{\mathcal{F}}(x) - \mathcal{F}_j(x)\| \leq \sum a_j \|\mathcal{F}_j(x) - \mathcal{F}'_j(x)\|. \tag{10}$$

*Thus expert evolution does not disrupt $\mathcal{H}$ and prevents catastrophic forgetting.*

**Theorem 3.2** (Convergence of Experts Evolution). *Under the elastic MoE adjustment mechanism in FedEMoE, the global model $M^{(t)}$ satisfies:*

$$\lim_{t \to \infty} \mathbb{E}\left[F(\theta^{(t)})\right] - F(\theta^*) \leq \mathcal{O}\left(\frac{1}{\sqrt{eT}}\right), \tag{11}$$

*where $\theta^*$ is the optimal parameter. The singular-value spectrum of $M^{(t)}$ remains bounded, and the expansion/shrinkage of experts does not violate the Lipschitz continuity of the loss landscape.*

### 3.5. Knowledge Exchange Mechanism

The sparse activation in MoE limits the Shared Expert (SE)'s exposure to full local data. To overcome this sparsity, we introduce a bidirectional knowledge exchange mechanism that ensures continuous knowledge flow between personalized and shared components, enhancing local personalization and collaborative learning.

**Enhancing SE with local personalized knowledge.** We enrich the SE with localized knowledge from the PEG. The purpose is to optimize some sub-experts in SE to make them more personalized. We measure the distribution mismatch by Kullback-Leibler (KL) divergence, defining the loss $\mathcal{L}_S$:

$$\mathcal{L}_S = \sum_{x \in \mathcal{D}_i} R_p(x) \log \frac{R_P(x)}{R_S(x)}, \tag{12}$$

where $R_S(x)$ is the SE output of sample $x$, and $R_P$ is the consensus output of all personalized experts in PEG. The consensus output fuses the personalized knowledge into a single target, which guides SE's output towards personalized

update direction. The consensus output $R_P$ is calculated as follows:

$$R_P(x) = \sum_{n=1}^{N_i} a_n R_n(x), \tag{13}$$

where $a_n$ represents the activation frequency of personalized expert $n$ in PEG, $R_n(x)$ is the output of personalized expert $n$ and $N_i$ is the number of personalized expert in $M_i$. This loss guides the SE to absorb and reflect the aggregated personalized knowledge.

**Enhancing local personality with SE.** Conversely, we optimize each personalized expert within the PEG to learn diverse, personalized patterns from the SE. This is achieved by minimizing a KL divergence loss $\mathcal{L}_p$ that pulls each expert's output towards the SE's output distribution:

$$\mathcal{L}_p = \sum_{n=1}^{N_i} \sum R_S(x) \log \frac{R_S(x)}{R_n(x)}. \tag{14}$$

This bidirectional exchange ensures that while the PEG deepens its personalization, it also remains informed by generally useful features captured by the SE, thereby enhancing both the specificity and robustness of the local model.

## 4. Experiments

We conduct extensive experiments to validate the effectiveness of the FedEMoE. Specifically, our experiments aim to address the following research questions: (RQ1) How does FedEMoE perform in heterogeneous-model settings? (RQ2) What is its effectiveness in homogeneous-model scenarios? (RQ3) How well does it handle diverse data distributions? (RQ4) How does our approach perform in terms of convergence speed? (RQ5) How scalable is FedEMoE? (RQ6) How does FedEMoE perform in terms of communication overhead? (RQ7) How much computation does each client incur per round? (RQ8) How sensitive is FedEMoE to key hyperparameters? (RQ9) How does each proposed component contribute to the overall performance?

### 4.1. Settings

**Dataset.** We test our approach FedEMoE on three datasets: CIFAR10, CIFAR100 (Krizhevsky et al., 2009), and TinyImageNet (Chrabaszcz et al., 2017). These datasets range from tasks and levels of difficulty, enabling a structured assessment of our approach.

**Baseline methods.** To evaluate FedEMoE, we focus on the statistical heterogeneity and model heterogeneity settings in federated learning and evaluate it with ten state-of-the-art baselines. Specifically, we include the classic homogeneous-model baseline FedAvg (McMahan et al., 2017); three personalization-oriented homogeneous-model methods, namely the adaptive local-aggregation

technique FedALA (Zhang et al., 2023c), the domain-bias-eliminating representation learning approach Fed-DBE (Zhang et al., 2023a), and the feature-alignment method FedPAC (Xu et al., 2023); the gradient-correction-based personalized algorithm FedGC (Niu & Deng, 2022); and five heterogeneous-model approaches that either distill knowledge (FedKD) (Wu et al., 2022), leverage class prototypes (FedProto,FedSA) (Tan et al., 2022; Zhou et al., 2025), combine meta-learning with representation learning (FedMRL) (Yi et al., 2024), and trainable global prototypes for better generalization (FedTGP) (Zhang et al., 2024).

**Statistical heterogeneity.** In line with prior research (Li et al., 2022; 2021), we introduce statistical heterogeneity among clients via the Dirichlet distribution (Hsu et al., 2019). The process involves sampling $q_{c,i}$ from $\text{Dir}(\beta)$ for each class $c$ and client $i$. Here, $\text{Dir}(\beta)$ denotes the Dirichlet distribution, with the $\beta$ set to 0.1 by default.

**Model heterogeneity.** In our experiments, model heterogeneity mainly consists of the following aspects: model basic architecture, the different number of personalized experts; the architecture of personalized experts. The model settings are kept consistent across all baselines. For detailed model architectures, refer to the Appendix E.2.

**Hyperparameter setting.** In our work, we explore complex scenarios in federated learning. We set the client participation rate of 0.2 per round. This setup allows us to evaluate the performance of various methods when only a small number of clients are active. We adopt one local training epoch on each client per round. We set the batch size at 64 and the learning rate $\eta$ at 0.005, spanning 100 communication rounds ($T$). Each client's local dataset is split into training (75%) and testing (25%) sets. We validate the personalized performance of the model on the local test sets. The server performs a scaling operation every $\mathcal{T}$ rounds ($\mathcal{T} = 20$). We use the Adam optimizer for local model updates. For all experiments, we run three trials and present the mean test accuracy. For detailed parameter settings, refer to the Appendix E.1.

### 4.2. Impact of Model Heterogeneity (RQ1)

We evaluate FedEMoE against the SOTA of HtFL methods. FedEMoE improves accuracy by 32.43% on CIFAR-100 and 40.62% on Tiny-ImageNet. Personalized expert groups learn client-specific multi-scale features, enriching local representations, while elastic shared experts provide a flexible mechanism for rich-knowledge transfer.

### 4.3. Impact of Model Homogeneity (RQ2)

To examine the performance of FedEMoE under model-homogeneous scenarios, we compare it with SOTA baselines. FedEMoE surpasses them by 22.78% on CIFAR-100

*Table 1.* Test accuracy(%) under different datasets in model-heterogeneous scenarios with K = 50.

| Dataset | CIFAR-10 | CIFAR-100 | Tiny-ImageNet |
|---|---|---|---|
| FedKD | 72.21 | 31.19 | 19.38 |
| FedProto | 68.83 | 25.13 | 2.57 |
| FedMRL | 78.37 | 36.01 | 19.38 |
| FedTGP | 66.47 | 31.96 | 12.21 |
| FedSA | 70.93 | 33.27 | 21.27 |
| Ours | 79.83 (↑1.9%) | 47.69 (↑32.4%) | 29.91 (↑40.6%) |

and 26.33% on Tiny-ImageNet. A decoupling splits personalized and shared parameters, letting each part learn on its own and removing mutual limits. Elastic shared experts expand the generalized knowledge pool and broaden universal representations without disturbing any client's personalized knowledge, so local decision boundaries stay intact.

*Table 2.* Test accuracy(%) under different datasets in model-homogeneous scenarios with K = 50.

| Dataset | CIFAR-10 | CIFAR-100 | Tiny-ImageNet |
|---|---|---|---|
| FedAvg | 76.85 | 35.94 | 23.68 |
| FedGC | 78.68 | 38.71 | 23.73 |
| FedALA | 79.01 | 30.83 | 19.03 |
| FedPAC | 75.21 | 20.82 | 19.81 |
| FedDBE | 75.85 | 36.16 | 20.92 |
| Ours | 79.22 (↑0.3%) | 47.53 (↑22.8%) | 29.98 (↑26.3%) |

### 4.4. Impact of Statistical Heterogeneity (RQ3)

In order to evaluate the performance of FedEMoE in handling statistical heterogeneity, we conducted a comparative study with other HtFL methods on CIFAR-100 and 50 clients. To control the level of statistical heterogeneity, we utilized the Dirichlet concentration parameter $\beta$, creating scenarios ranging from highly imbalanced data ($\beta = 0.1$) to relatively balanced data distribution ($\beta = 10$). The comparative results are presented in Figure 3. FedEMoE consistently outperforms baselines across all heterogeneity levels, with performance gaps widening under higher imbalance, demonstrating superior robustness to non-IID data distributions.

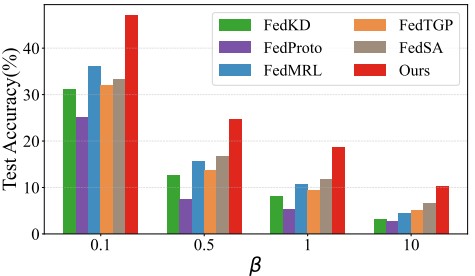

*Figure 3.* Test accuracy of different methods under diverse statistical heterogeneity.

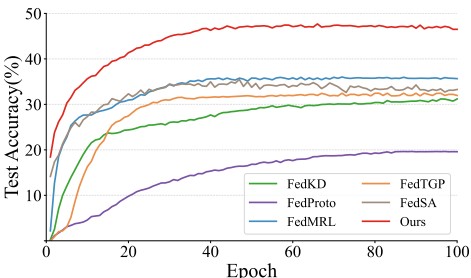

*Figure 4.* The test accuracy curves of different methods under heterogeneous settings.

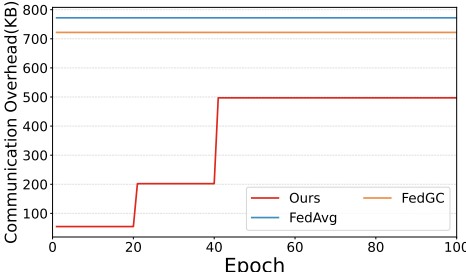

*Figure 5.* The communication overhead of different methods.

### 4.5. Convergence Speed Analysis (RQ4)

To evaluate the convergence speed of our approach, we conducted experiments testing SOTA methods on CIFAR-100 with 50 clients. As shown in Figure 4, FedEMoE significantly outperforms others in both convergence speed and peak test accuracy. FedEMoE shows no degradation in generalization and exhibits no evident model drift, demonstrating that the proposed decoupling of personalization and generalization is indeed effective.

### 4.6. Scalability analysis (RQ5)

To evaluate the scalability of our approach, we conducted experiments testing SOTA methods across varying numbers of clients. As shown in Table 3, our approach outperformed other methods, particularly on the CIFAR-100 and Tiny-ImageNet datasets. In scenarios with 100 clients, it achieved remarkable accuracy improvements of 18% and 48%, respectively.

On CIFAR-100, increasing clients shrinks each local set to about 500 images. Consequently, PEG experts begin to over-fit, narrowing our lead to 18%. On Tiny-ImageNet, the richer data let the elastic SE continuously spawn new specialists, widening the gap to 48%. Thanks to the elastic strategy, our approach can effectively handle scenarios with a large number of clients while maintaining generalized performance, which is crucial for real-world applications.

### 4.7. Communication Overhead Analysis (RQ6)

Figure 5 shows FedEMoE's communication overhead compared to model-based federated learning methods. Despite the expanding shared-expert mechanism, our peak consumption remains lower than all baselines, as we upload only the lightweight and sparse SE instead of the entire model. Communication overhead stops increasing once the model converges, confirming the effectiveness of our expert status diagnosis and expert evolution.

### 4.8. Client's Resource Consumption (RQ7)

Table 4 reports the per-client computational overhead throughout per round. For FedEMoE, each client performs only standard local model updates and logit-level knowledge distillation. FedEMoE incurs approximately 1.24 GFLOPs of extra computation per client, which is moderate compared to FedKD (2.61 GFLOPs) and FedMRL (1.31 GFLOPs), while being higher than FedProto (0.417 GFLOPs) and FedTGP (0.424 GFLOPs). This additional cost primarily stems from the auxiliary distillation loss on logits. Crucially, all resource-intensive operations are offloaded to the server, ensuring that the per-client computational burden stays feasible for practical deployment.

### 4.9. Sensitivity Analysis (RQ8)

To investigate the impact of the interval for shared expert scaling operations on the server, we evaluate model accuracy under Dirichlet $\beta = 0.1$. As illustrated in Figure 6, an interval of 20 iterations yields the best performance. We attribute this phenomenon to two complementary factors. First, overly frequent ($\mathcal{T} = 10$) scaling operations continuously perturb the model architecture, impeding sufficient convergence. Conversely, excessively sparse ($\mathcal{T} = 30, 40, 50$) operations curtail structural adaptation, preventing the shared experts from evolving toward an optimal configuration. We provide more sensitivity experiments in the Appendix F.4.

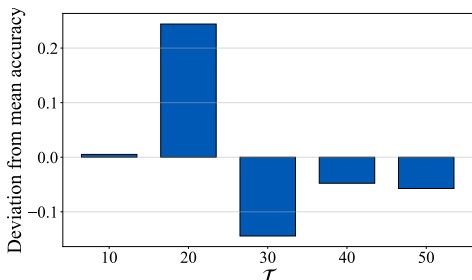

*Figure 6.* The deviation from average accuracy on different intervals of elastic operation.

In Table 5 we co-vary the number of shared experts N (3, 4, 5) and the gate sparsity topK (1, 2, 3). All runs use $\mathcal{T} = 2$.

*Table 3.* Test accuracy(%) with different client numbers on datasets in HtFL scenarios.

| Dataset | CIFAR-100 | | | Tiny-ImageNet | | |
|---|---|---|---|---|---|---|
| K | 20 Clients | 50 Clients | 100 Clients | 20 Clients | 50 Clients | 100 Clients |
| FedKD | 33.62 | 31.19 | 29.41 | 18.24 | 19.38 | 17.44 |
| FedProto | 29.12 | 25.13 | 23.11 | 6.93 | 2.57 | 1.51 |
| FedMRL | 39.19 | 36.01 | 35.52 | 18.74 | 19.38 | 17.21 |
| FedTGP | 35.64 | 31.96 | 35.19 | 12.61 | 12.21 | 11.33 |
| FedSA | 39.41 | 33.27 | 31.65 | 28.18 | 21.27 | 17.43 |
| **Ours** | **49.03(↑24.41%)** | **47.69(↑32.43%)** | **41.76(↑18.67%)** | **29.12(↑3.33%)** | **29.91(↑40.62%)** | **25.82(↑48.05%)** |

*Table 4.* Per-client computational overhead on CIFAR-100.

| Method | Total Time (s) | Local extra GFLOPs |
|---|---|---|
| FedKD | 352 | 2.61 |
| FedProto | 105 | 0.417 |
| FedMRL | 91 | 1.31 |
| FedTGP | 145 | 0.424 |
| FedSA | 112 | 0.426 |
| **Ours** | **93** | **1.24** |

The best result (49.03%) is again obtained with N = 3 and topK = 2. Adding more experts or activating more of them increases communication consumption and encourages each expert to memorize tiny client-specific patterns, which hurts generalization. Restricting the gate to topK = 1, on the other hand, under-uses the ensemble and drops accuracy by roughly 0.4%. The small spread (less than 0.8%) indicates robustness to this design choice.

*Table 5.* Joint sensitivity to expert count($N$) and topK.

| topK \ experts | 3 | 4 | 5 |
|---|---|---|---|
| 1 | 49.60 | 47.46 | 46.99 |
| 2 | **49.03** | 47.33 | 46.88 |
| 3 | 48.80 | 48.62 | 46.31 |

Table 6 tests the expansion trigger. We sweep the rank threshold $v_0 \in (0.75, 0.85, 0.95)$ and the exponential-smoothing factor $\gamma \in (0.05, 0.07, 0.10)$, keeping $\mathcal{T} = 2$, N = 3 and $topK = 2$ fixed. The pair ($v_0 = 0.85$, $\gamma = 0.07$) gives the highest accuracy (49.03%). A lower threshold splits experts too early and bloats the model, whereas a higher one delays growth and leaves under-fitting unchanged. $\gamma = 0.07$ strikes a balance: it smooths mini-batch noise yet still reacts within a few rounds. The accuracy variation across the nine settings is below 1%, confirming mild sensitivity.

### 4.10. Ablation Studies (RQ9)

In the Table 7, the results show that activating all three components: personalized experts (PEG), shared experts (SE)

*Table 6.* Sensitivity to expansion threshold $v_0$ and EMA factor $\gamma$.

| $\gamma \backslash v_0$ | 0.75 | 0.85 | 0.95 |
|---|---|---|---|
| 0.05 | 48.11 | 48.32 | 48.35 |
| 0.07 | 48.28 | **49.03** | 47.26 |
| 0.10 | 48.40 | 49.09 | 47.61 |

with EMoE architecture, and knowledge exchange, yields 47.69% accuracy. Disabling knowledge exchange drops the score to 43.85%, confirming that multi-scale feature is indispensable. Retaining only PEG and knowledge exchange while removing SE's EMoE architecture causes a sharp decline to 42.25%, underscoring the role of SE in transferring knowledge and flexible fitting. Conversely, keeping SE and knowledge exchange but omitting PEG's multi-scale extraction achieves 35.46%, highlighting the critical contribution of PEG. Collectively, the three modules are all essential for optimal performance.

*Table 7.* Ablation studies on the key components of FedEMoE on CIFAR-100 with K = 50.

| Variant | Accuracy(%) |
|---|---|
| Ours | 47.69 |
| Ours(w/o Knowledge Exchange) | 43.85(-3.84) |
| Ours(w/o PEG) | 35.46(-12.23) |
| Ours(w/o SE) | 42.25(-5.44) |

## 5. Conclusion

In this paper, we propose FedEMoE to decouple personalization from generalization on HtFL. FedEMoE designs a multi-scale knowledge extraction and a knowledge exchange mechanism to break the bottleneck of knowledge transfer. Moreover, FedEMoE first employs an EMoE model architecture to highlight multiple personalized knowledge. Extensive experiments show that FedEMoE surpasses existing HtFL methods under diverse statistical and model heterogeneity settings.

## Impact Statement

This paper presents work whose goal is to advance the field of Machine Learning. There are many potential societal consequences of our work, none which we feel must be specifically highlighted here.

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

## A. The Use of Large Language Models

In the paper writing stage of this study, we only utilized Large Language Models (LLMs) as a general language polishing tool. Specifically, we employed LLMs to conduct spelling checks, grammatical corrections, and word optimization on certain sentences in the text, aiming to enhance the clarity and fluency of language expression. LLMs did not participate in the proposal of research questions, experimental design, data analysis, result interpretation, chart production, or the generation of any scientific content. Therefore, the role of LLMs in this work is limited to assisting in language-level polishing and does not constitute an academic contributor.

## B. Notations

*Table 8.* Notations

| Symbol | Description |
|---|---|
| $K$ | Total number of clients. |
| $i$ | The client number. |
| $p$ | client participation fraction. |
| $T$ | Total global training rounds. |
| $\mathcal{T}$ | shared expert scaling interval. |
| $D_i$ | The client $i$'s private dataset. |
| $\mathcal{D}_i$ | The client $i$'s training dataset. |
| $M_i$ | The model parameter of the $i$-th client. |
| $C$ | Number of classes. |
| $\mathcal{N}_i$ | The number of experts in $M_i$. |
| $N$ | The number of experts in shared expert. |
| Top $k$ | The number of actived experts. |
| $FE_i$ | The feature extractor of client $i$. |
| $SE_i$ | The shared expert of client $i$. |
| $PEG_i$ | The personalized expert group of client $i$. |
| $H_i$ | The personalized header of client $i$. |
| $f_i(\cdot)$ | $FE_i$'s parameter. |
| $p_i(\cdot)$ | $PEG_i$'s parameter. |
| $s_i(\cdot)$ | $SE_i$'s parameter. |
| $h_i(\cdot)$ | $H_i$'s parameter. |

## C. Algorithm

**Algorithm 4** ClientUpdate

**Input:** Global feature extractor $f_g^{t-1}$, Global shared expert $s_g^{t-1}$, Client model $M_i^{t-1}$, Local training data $\mathcal{D}_i$
**Output:** Feature extractor $f_i^t$, Shared expert $s_i^t$
 1: $M_i^{t-1} \leftarrow \text{UpdateLocalModel}(f_g^{t-1}, s_g^{t-1}, M_i^{t-1})$
 2: DoubleKD $(M_i^{t-1}, \mathcal{D}_i)$ via equation 12 and equation 13
 3: $M_i^t \leftarrow \text{LocalTrain}(M_i^{t-1}, \mathcal{D}_i)$
     $f_i^t, s_i^t$

## D. Theoretical Analysis

### D.1. The Expert Initialization in Expansion.

When migrating expert network $e_j$ from the original MoE to a larger-scale new MoE $e^*$, we can simply keep the $j$-th expert unchanged by setting $e_j^{new} = e_j^{old}$, while initializing all remaining positions $i \neq j$, where $j$ is the index of $e_j$ in the old MoE $W_g^{old}$. This design is justified by two key properties. First, the routing distribution remains invariant:

the original gate's preference for the $j$-th expert is expressed as $p_j(x) = exp(h^T w_j^{old})/\sum_k exp(h^T w_k^{old})$. By this, we ensure $p_j^{new}(x) \approx p_j^{old}(x)$ for any input $x$, thereby preventing catastrophic forgetting. Second, the overall output of the new MoE can be written as $y^{new}(x) = G^{new}(x)je_j^{old}(x) + \sum_{i \neq j} G^{new}(x)_i e_i^{new}(x)$. The first term preserves existing capability, while the second term, thanks to the random initialization of $E_i^{new}$, forms an approximately orthogonal basis that introduces additional representational freedom. During training, the gate will sparsely assign new tokens to these fresh experts, delivering a provably incremental expansion of model capacity without erasing prior knowledge.

## D.2. Global Check via Singular Spectrum

**Theorem D.1.** *Let round $t$ proceed solely by server-side weighted aggregation of the uploaded local models $M_i$ into a global model $M^{(t)} = \sum_{i=1}^p w_i M_i$, with $\sum_i w_i = 1$, $w_i \geq 0$. Define the per-round client covariance $\Sigma = \sum_i w_i (M_i - M)(M_i - M)^T$. Then, for every k, with probability at least $1 - \frac{1}{poly(d)}$, we have:*

$$|\sigma_k(M^{(t)}) - \sigma_k(M)| \leq \sqrt{\frac{||\Sigma||_2 logd}{p}} + \frac{Llogd}{3p}, \tag{15}$$

*where $\sigma_k(\cdot)$ denotes the k-th largest singular value, L is the Lipschitz constant of the model parameters, and p is the number of participating clients, $poly(d)$ is a notation emphasizing that the probability approaches 1 exponentially as the dimension d increases.*

**Remark 1**. The Theorem D.1 justifies using the aggregated model's weights as a proxy for the training status of all participating models. If the singular spectrum of $M^{(t)}$ decays rapidly (light tail), $||\Sigma||_2$ must be a small value, indicating negligible drift and confirming that the aggregated model is already close to the ideal drift-free model $M_i$. Conversely, a heavy tail or abrupt large singular values directly reveal substantial drift, flagging the model as unreasonable. Thus, the aggregated weight matrix alone suffices for a theoretically grounded validity check.

## D.3. Convergence Analysis for Strongly Convex Case

In this subsection, we provide a formal convergence guarantee for the proposed FedEMoE approach under standard assumptions commonly used in federated optimization.

### D.3.1. ASSUMPTION

**Assumption D.2** (Smoothness). $F$ is $L$-smooth: for all vector $\theta_1$ and $\theta_2$,

$$||\nabla F(\theta_1) - \nabla F(\theta_2)|| \leq L||\theta_1 - \theta_2||.$$

Another form:

$$F(\theta_1) \leq F(\theta_2) + <\theta_1 - \theta_2, \nabla F(\theta_2)> + \frac{L}{2}||\theta_1 - \theta_2||^2.$$

**Assumption D.3** (Convexity). Assume $F$ is strongly convex with parameter $\mu$. For any vector $\theta_1, \theta_2$, we have:

$$F(\theta_1) \geq F(\theta_2) + <\theta_1 - \theta_2, \nabla F(\theta_2)> + \frac{\mu}{2}||\theta_1 - \theta_2||^2.$$

**Assumption D.4** (Stochastic Gradient Variance Bounded).

$$E[||\nabla F(\theta; x, y) - \nabla F(\theta)^2] \leq \sigma^2.$$

**Assumption D.5** (MoE Boundedness). The gating weights $\{\pi_j\}$ satisfy $\sum_{j=1}^k \pi_j$ and each expert's outputs and gradients are uniformly bounded.

### D.3.2. KEY LEMMAS

**Lemma D.6.** *Assume Assumption D.2, D.3, D.4 hold, if local learning rate $\eta_l \leq \frac{1}{L}$. Then the local update on the client side satisfies:*

$$E||\theta_{i,E}^t|| \leq (1 - \eta_l \mu)^e ||\theta^t - \theta^*|| + \frac{2\eta_l \sigma^2}{\eta}. \tag{16}$$

**Lemma D.7.** *Assume Assumption D.2, D.3 hold. Then the aggregation on the server satisfies:*

$$E||\theta^{t+1} - \theta^*||^2 \leq \sum_i w_i E||\theta_{i,e}^t - \theta^*||^2. \tag{17}$$

**Lemma D.8.** *Set linear projection P satisfies $||p|| \leq 1$ then*

$$||P(x) - \theta^*|| \leq ||x - \theta^*||. \tag{18}$$

### D.3.3. PROOF OF KEY LEMMA

*Proof.* (Proof of Lemma 1) For any local step $\gamma$ Taking squared norms and expectations,

$$\Delta_{\gamma+1} = \Delta_\gamma - \eta_l \nabla f(\theta_{i,\gamma}^t; \xi_{i,\gamma}^t). \tag{19}$$

Taking squared norms and expectations, we can get

$$
\begin{aligned}
E&||\Delta_{\gamma+1}||^2 \\
=&E||\Delta_\gamma - \eta_l \nabla f(\theta_{i,\gamma}^t; \xi_{i,\gamma}^t)||^2. \\
=&E[||\Delta_\gamma||^2 - 2\eta_l <\Delta_\gamma, \nabla f(\theta_{i,\gamma}^t; \xi_{i,\gamma}^t> +\eta_l^2 ||\nabla f(\theta_{i,\gamma}^t; \xi_{i,\gamma}^t||^2]
\end{aligned} \tag{20}
$$

Substituting assumption D.4 yields:

$$E||\Delta_{\gamma+1}||^2 \leq ||\Delta_\gamma||^2 - 2\eta_l <\Delta_\gamma, \nabla F(\theta_{i,\gamma}^t) > +\eta_l^2(||\nabla F(\theta_{i,\gamma}^t)||^2 + \sigma^2) \tag{21}$$

Substituting assumption D.3,D.2 yields:

$$E||\Delta_{\gamma+1}||^2 \leq (1 - \eta_l \mu)||\Delta_\gamma||^2 + \eta_l^2 \sigma^2 \tag{22}$$

Unrolling the recursion for $\sigma = 0, 1, \ldots, e-1$:

$$E||\Delta_e||^2 \leq (1 - \mu\eta_l)^e ||\Delta_0||^2 + \eta_l^2 \sigma^2 \sum_{\gamma=0}^{e-1}(1 - \mu\eta_l)^\gamma. \tag{23}$$

Therefore,

$$E||\Delta_e||^2 \leq (1 - \mu\eta_l)^e ||\theta^t - \theta^*||^2 + \frac{\eta_l \sigma^2}{\mu}. \tag{24}$$

To match the statement in Lemma D.6, we further loosen the constant term to absorb higher-order effects:

$$E||\theta_{i,E}^t|| \leq (1 - \eta_l \mu)^e ||\theta^t - \theta^*|| + \frac{2\eta_l \sigma^2}{\eta}. \tag{25}$$

$$\square$$

*Proof.* (Proof of Lemma 2) From Lemma D.6 (already proved) we have for every client i:

$$E||\theta_{i,E}^t - \theta^*||^2 \leq (1 - \mu\eta_l)^e ||\theta_g^t - \theta^*||^2 + \frac{2\eta_l \sigma^2}{\mu}. \tag{26}$$

According to Jensen's inequality (convex combination), we can get:

$$
\begin{aligned}
E&\left|\left|\sum_{i=1}^N w_i \theta_{i,e}^t - \theta^*\right|\right|^2 \\
\leq& \sum_{i=1}^N w_i E||\theta_{i,e}^t - \theta^*||^2 \\
\leq& (1 - \mu\eta_l)^e ||\theta_g^t - \theta^*||^2 + \frac{2\eta_l \sigma^2}{\mu}.
\end{aligned} \tag{27}
$$

Because $P$ is non-expansive and $\theta^*$ is in the fixed-point set of $P$

$$\|\theta_g^{t+1} - \theta^*\| = \|P(\bar{\theta}^{t+1}) - \theta^*\| \leq \|\bar{\theta}^{t+1} - \theta^*\|, \tag{28}$$

where $\bar{\theta}^{t+1} = \sum_i w_i \theta_{i,e}^t$. Substituting assumption D.3,D.2 yields:

$$F(\theta_{\text{virt}}) \leq F(\theta_g^t) - \eta_g \left(1 - \frac{L\eta_g}{2}\right) 2\mu\big(F(\theta_g^t) - F(\theta^*)\big). \tag{29}$$

With $\eta_g \leq 2\mu^2/\beta^3$ (choose $\beta = 2L$ for simplicity),so

$$F(\theta_{\text{virt}}) \leq F(\theta_g^t) - \mu\eta_g\big(F(\theta_g^t) - F(\theta^*)\big) + \frac{\eta_g^2 \beta \sigma^2}{2}. \tag{30}$$

Collecting all residual terms and using the previous bounds, we obtain:

$$E[L_a(\theta_g^{t+1})] \leq E[L_a(\theta_g^t)] - \frac{\mu^2}{2}\eta_g E\|\theta_g^t - \theta^*\|^2 + \frac{\eta_g^2 \beta \sigma^2}{2}. \tag{31}$$

$\square$

*Proof.* (Proof of Lemma 3) Define the error vector $\Delta_t = \tilde{\theta}_i^t - \tilde{\theta}_i^*$.. The update gives $\Delta_{t+1} = \Delta_t - \rho\nabla f_i(\tilde{\theta}_i^t; \xi_i^t)$. Taking squared norms and expectations:

$$E\|\Delta_{t+1}\|^2 = E\|\Delta_t - \rho\nabla f_i(\tilde{\theta}_i^t; \xi_i^t)\|^2 \tag{32}$$
$$= E\Big[\|\Delta_t\|^2 - 2\rho\langle\Delta_t, \nabla f_i(\tilde{\theta}_i^t; \xi_i^t)\rangle + \rho^2\|\nabla f_i(\tilde{\theta}_i^t; \xi_i^t)\|^2\Big].$$

Substituting assumption D.3,D.2 yields:

$$E\|\Delta_{t+1}\|^2 \leq \|\Delta_t\|^2 + 2\rho\Big(F_i(\tilde{\theta}_i^t) - F_i(\tilde{\theta}_i^*) - \frac{\mu}{2}\|\Delta_t\|^2\Big) + \rho^2\Big(\gamma\big(F_i(\tilde{\theta}_i^t) - F_i(\tilde{\theta}_i^*)\big) + \sigma^2\Big) \tag{33}$$
$$= (1 - \mu\rho)\|\Delta_t\|^2 + \big(2\rho + \rho^2\gamma\big)\big(F_i(\tilde{\theta}_i^t) - F_i(\tilde{\theta}_i^*)\big) + \rho^2\sigma^2.$$

Because $\rho \leq 2/\gamma$ implies $2\rho + \rho^2\gamma \leq 0$, and $F_i(\dots) - F_i(\dots^*) \leq 0$, we obtain:

$$E\|\Delta_{t+1}\|^2 \leq (1 - \mu\rho)\|\Delta_t\|^2 + \rho^2\sigma^2. \tag{34}$$

Substituting assumption D.4 yields:

$$E[L_p(\tilde{\theta}_i^{t+1})] \leq E[L_p(\tilde{\theta}_i^t)] - \mu^2\rho E\|\tilde{\theta}_i^t - \tilde{\theta}_i^*\|^2 + \frac{\gamma\rho^2\sigma^2}{2}. \tag{35}$$

Lemma D.8 is proved, showing that each personalized-parameter update yields a linear decrease in the expected personalized loss, up to a variance term controlled by rho

$\square$

### D.3.4. MAIN THEOREM

**Theorem D.9.** *Suppose Assumptions D.2,D.3,D.4 and D.5 hold. Let the local learning rate satisfy $\eta_l \leq \min\left\{\frac{1}{L}, \frac{\log(e)}{e\mu}\right\}$,. Then:*

$$E[F(\theta_{\text{final}}^T) - F(\theta^*)] \leq \frac{2L\sigma^2}{\mu^2 eT} + \frac{\mu}{2}\exp(-\mu\eta_l eT)\|\theta^0 - \theta^*\|^2. \tag{36}$$

*According to Lemma 1,2,3, we can get:*

$$\sum_{k=0}^{T-1}(1 - \mu\eta_l)^{ek} \leq \sum_{k=0}^{\infty}(1 - \mu\eta_l)^{ek} = \frac{1}{1 - (1 - \mu\eta_l)^e} \leq \frac{1}{\mu\eta_l e}, \tag{37}$$

*because $(1-x)^e \leq 1 - ex$ for $0 < x < 1/e$, hence $D_T \leq (1-\mu\eta_l)^{eT} D_0 + \frac{2\sigma^2}{\mu^2 e}$. By strong convexity, $F(\theta) - F(\theta^*) \leq \frac{L}{2}\|\theta - \theta^*\|^2$. Apply to $\theta = \theta_{final}^T = P(\theta_g^T)$:*

$$E[F(\theta_{final}^T) - F(\theta^*)] \leq \frac{L}{2} E\|\theta_g^T - \theta^*\|^2 \leq \frac{L}{2}\left[(1-\mu\eta_l)^{eT} D_0 + \frac{2\sigma^2}{\mu^2 e}\right]. \tag{38}$$

*Take $\eta_l = \frac{\log(T)}{e\mu T}$. Then*

$$(1-\mu\eta_l)^{eT} \leq \exp(-\mu\eta_l eT) = \exp(-\log T) = \frac{1}{T}. \tag{39}$$

*Thus:*

$$E[F(\theta_{final}^T) - F(\theta^*)] \leq \frac{L}{2}\left[\frac{D_0}{T} + \frac{2\sigma^2}{\mu^2 e}\right]. \tag{40}$$

*FedEMoE converges at a rate $\mathcal{O}\left(\frac{1}{T} + \frac{1}{e}\right)$, showing linear speed-up with the number of local steps $e$ and communication rounds $T$.*

### D.4. Convergence Analysis for Non-convex Case

**Assumption D.10** (Smoothness). $F$ is $L$-smooth: for all vector $\theta_1$ and $\theta_2$,

$$\|\nabla F(\theta_1) - \nabla F(\theta_2)\| \leq L\|\theta_1 - \theta_2\|. \tag{41}$$

**Assumption D.11** (Stochastic Gradient Variance Bounded).

$$E[\|\nabla F(\theta; x, y) - \nabla F(\theta)^2] \leq \sigma^2. \tag{42}$$

**Assumption D.12** (MoE Boundedness).

$$|\mathbf{e}_j(x)| \leq G, \ |\nabla\mathbf{e}_j(x)| \leq L_e \tag{43}$$

#### D.4.1. KEY LEMMA

**Lemma D.13.** *If the local learning rate satisfies $\eta_l \leq \frac{1}{2L}$, then:*

$$\frac{1}{N}\sum_{i=1}^N E\|x_{i,e}^t - x^t\|^2 \leq 4e^2\eta_l^2\|\nabla F(x^t)\|^2 + 2e\eta_l^2\sigma^2. \tag{44}$$

**Lemma D.14.** *For a non-expansive linear projection $P$ with operator norm $\|P\| \leq 1$, the aggregated update satisfies:*

$$E\|x^{t+1} - x^t\|^2 \leq \frac{1}{N}\sum_{i=1}^N E\|x_{i,e}^t - x^t\|^2. \tag{45}$$

**Lemma D.15.** *If the local step size satisfies $\eta_l \leq \frac{1}{2L}$, then:*

$$E[F(x^{t+1})] \leq E[F(x^t)] - \frac{e\eta_l}{4}E\|\nabla F(x^t)\|^2 + \frac{Le\eta_l^2\sigma^2}{2}. \tag{46}$$

#### D.4.2. PROOF OF KEY LEMMA

*Proof.* (Proof of Lemma D.13) Take squared norms and expectations:

$$E\|\delta_{i,\tau+1}\|^2 = E\|\delta_{i,\tau} - \eta_l g_{i,\tau}^t\|^2 = E\|\delta_{i,\tau}\|^2 - 2\eta_l E\langle\delta_{i,\tau}, g_{i,\tau}^t\rangle + \eta_l^2 E\|g_{i,\tau}^t\|^2. \tag{47}$$

Substituting assumption D.11,D.10 yields:

$$E\|g_{i,\tau}^t\|^2 \leq 2\|\nabla F_i(x^t)\|^2 + 2L^2\|\delta_{i,\tau}\|^2 + \sigma^2. \tag{48}$$

Insert the bound, we can get:

$$E\|\delta_{i,\tau+1}\|^2 \leq E\|\delta_{i,\tau}\|^2 - 2\eta_l E\langle\delta_{i,\tau}, \nabla F_i(x^t)\rangle + \eta_l^2\left[2\|\nabla F_i(x^t)\|^2 + 2L^2\|\delta_{i,\tau}\|^2 + \sigma^2\right]. \tag{49}$$

Then:

$$-2\eta_l E\langle\delta_{i,\tau}, \nabla F_i(x^t)\rangle \leq 2\eta_l^2\|\nabla F_i(x^t)\|^2 + \frac{1}{2}\|\delta_{i,\tau}\|^2. \tag{50}$$

Unrolling over $\tau = 0, \ldots, e-1$ gives:

$$E\|\delta_{i,e}\|^2 \leq 4e^2\eta_l^2\|\nabla F_i(x^t)\|^2 + 2e\eta_l^2\sigma^2. \tag{51}$$

Taking the average over all clients:

$$\frac{1}{N}\sum_{i=1}^N E\|\delta_{i,e}\|^2 \leq 4e^2\eta_l^2\|\nabla F(x^t)\|^2 + 2e\eta_l^2\sigma^2, \tag{52}$$

$$\square$$

*Proof.* (Proof od Lemma D.14) The squared norm is convex, so by Jensen's inequality$\bar{x}^{t+1} = \sum_{i=1}^N w_i x_{i,e}^t$. The squared norm is convex, so by Jensen's inequality:

$$\|\bar{x}^{t+1} - x^t\|^2 = \left\|\sum_{i=1}^N w_i(x_{i,e}^t - x^t)\right\|^2 \leq \sum_{i=1}^N w_i\|x_{i,e}^t - x^t\|^2. \tag{53}$$

Taking expectation over the randomness yields

$$E\|\bar{x}^{t+1} - x^t\|^2 \leq \sum_{i=1}^N w_i E\|x_{i,e}^t - x^t\|^2 = \frac{1}{N}\sum_{i=1}^N E\|x_{i,e}^t - x^t\|^2. \tag{54}$$

The final update is $x^{t+1} = P(\bar{x}^{t+1})$, where the linear operator $P$ satisfies$\|P(u) - P(v)\| \leq \|u - v\|, \quad \forall u, v$. Hence:

$$\|x^{t+1} - x^t\|^2 = \|P(\bar{x}^{t+1}) - P(x^t)\|^2 \leq \|\bar{x}^{t+1} - x^t\|^2. \tag{55}$$

Combining the above gives the desired bound:

$$E\|x^{t+1} - x^t\|^2 \leq \frac{1}{N}\sum_{i=1}^N E\|x_{i,e}^t - x^t\|^2. \tag{56}$$

$$\square$$

*Proof.* (Proof of Lemma D.15) By Assumption D.10, for any two points we have $F(y) \leq F(x) + \langle\nabla F(x), y - x\rangle + \frac{L}{2}\|y - x\|^2$. Set $x = x^t, \quad y = x^{t+1}$. Then:

$$E[F(x^{t+1})] \leq E[F(x^t)] + E\langle\nabla F(x^t), x^{t+1} - x^t\rangle + \frac{L}{2}E\|x^{t+1} - x^t\|^2. \tag{57}$$

From Lemma D.14 we have:

$$E\|x^{t+1} - x^t\|^2 \leq \frac{1}{N}\sum_{i=1}^N E\|x_{i,e}^t - x^t\|^2. \tag{58}$$

Write the inner product term explicitly:

$$E\langle\nabla F(x^t), x^{t+1} - x^t\rangle = E\left\langle\nabla F(x^t), \sum_{i=1}^N w_i(x_{i,e}^t - x^t)\right\rangle = \sum_{i=1}^N w_i E\langle\nabla F(x^t), x_{i,e}^t - x^t\rangle. \tag{59}$$

For each client $i$ and step $\tau$ we have:

$$x_{i,e}^t - x^t = -\eta_l \sum_{\tau=0}^{e-1} g_{i,\tau}^t, \tag{60}$$

so we can get:

$$E\langle \nabla F(x^t), x_{i,e}^t - x^t \rangle \tag{61}$$

$$= -e\eta_l \langle \nabla F(x^t), \nabla F_i(x^t) \rangle \tag{62}$$

$$\leq -e\eta_l \|\nabla F(x^t)\|^2 + 2Le^2\eta_l^3 \|\nabla F(x^t)\|^2 + e\eta_l^2 \sigma^2. \tag{63}$$

Insert the bounds from all above:

$$E[F(x^{t+1})] \leq E[F(x^t)] - e\eta_l \|\nabla F(x^t)\|^2 + 2Le^2\eta_l^3 \|\nabla F(x^t)\| \tag{64}$$

$$+ e\eta_l^2 \sigma^2 + \frac{L}{2}\left(4e^2\eta_l^2 \|\nabla F(x^t)\|^2 + 2e\eta_l^2 \sigma^2\right).$$

When $\eta_l \leq 1/(2L)$ we have $2Le^2\eta_l^3 + 2Le^2\eta_l^2 \leq \frac{e\eta_l}{4}$, and $e\eta_l^2\sigma^2 + Le\eta_l^2\sigma^2 \leq \frac{Le\eta_l^2\sigma^2}{2}$. Hence:

$$E[F(x^{t+1})] \leq E[F(x^t)] - \frac{e\eta_l}{4} E\|\nabla F(x^t)\|^2 + \frac{Le\eta_l^2\sigma^2}{2}. \tag{65}$$

$\square$

### D.4.3. MAIN THEOREM FOR NON-CONVEX CASE

**Theorem D.16.** *Choose the local learning rate $\eta_l = \frac{1}{\sqrt{eTL}}$. Then after $T$ communication rounds,*

$$\frac{1}{T}\sum_{t=0}^{T-1} E\|\nabla F(x^t)\|^2 \leq \frac{2L\Delta_0}{\sqrt{eT}} + \frac{\sigma^2}{\sqrt{eT}}, \tag{66}$$

*We have for each round $t$:*

$$E[F(x^{t+1})] \leq E[F(x^t)] - \frac{e\eta_l}{4} E\|\nabla F(x^t)\|^2 + \frac{Le\eta_l^2\sigma^2}{2}. \tag{67}$$

*Move the gradient term to the left, we can get:*

$$\frac{e\eta_l}{4} E\|\nabla F(x^t)\|^2 \leq E[F(x^t)] - E[F(x^{t+1})] + \frac{Le\eta_l^2\sigma^2}{2}. \tag{68}$$

*Sum the above inequality over $t = 0, 1, \ldots, T-1$:*

$$\sum_{t=0}^{T-1} \frac{e\eta_l}{4} E\|\nabla F(x^t)\|^2 \leq F(x^0) - E[F(x^T)] + \frac{Le\eta_l^2\sigma^2 T}{2}. \tag{69}$$

*Because $F(x^0) - E[F(x^T)] \leq F(x^0) - F = \Delta_0$. Hence:*

$$\sum_{t=0}^{T-1} \frac{e\eta_l}{4} E\|\nabla F(x^t)\|^2 \leq \Delta_0 + \frac{Le\eta_l^2\sigma^2 T}{2}. \tag{70}$$

*Insert $\eta_l = \frac{1}{\sqrt{eTL}}$, we can get:*

$$\frac{\sqrt{e}}{4\sqrt{TL}} \sum_{t=0}^{T-1} E\|\nabla F(x^t)\|^2 \leq \Delta_0 + \frac{\sigma^2}{2}. \tag{71}$$

*Then:*

$$\frac{1}{T}\sum_{t=0}^{T-1} E\|\nabla F(x^t)\|^2 \leq \frac{4\sqrt{TL}}{\sqrt{eT}}\Delta_0 + \frac{4\sqrt{TL}}{\sqrt{eT}}\frac{\sigma^2}{2} = \frac{4\sqrt{L}}{\sqrt{eT}}\Delta_0 + \frac{2\sqrt{L}}{\sqrt{eT}}\sigma^2. \tag{72}$$

*Under non-convexity, FedEMoE achieves $\mathcal{O}\left(\frac{1}{\sqrt{eT}}\right)$ convergence to a stationary point, linearly dependent on the product of local steps $e$ and communication rounds $T$.*

## D.5. Proof of Convergence of Expansion and Shrinkage

### D.5.1. EXPANSION DOES NOT INCREASE THE LIPSCHITZ CONSTANT

**Lemma D.17.** *Let the original model parameter be $\theta \in \mathbb{R}^d$, and after applying the **expansion** operation, the new parameter becomes:*

$$\theta' = (e_1, e_2, \ldots, e_N) \in \mathbb{R}^{N \cdot d} \quad with \quad e_p = e, \quad and\ other\ e_i\ initialized\ arbitrarily.$$

*We assume the loss function $F_k(\theta)$ is **Lipschitz continuous** in $\theta$, i.e.,*

$$\|\nabla F_k(\theta_1) - \nabla F_k(\theta_2)\| \le L\|\theta_1 - \theta_2\|$$

*Proof.* Let us denote:

$$\theta'_1 = (e_{1,1}, \ldots, e_{1,N}), \quad \theta'_2 = (e_{2,1}, \ldots, e_{2,N}) \tag{73}$$

Since **expansion** only introduces new parameters and does **not modify** the original expert $e$, we can write:

$$\nabla_{\theta'} F(\theta') = (\nabla_{e_1} F(\theta'), \ldots, \nabla_{e_N} F(\theta')) \tag{74}$$

Note that:

- $\nabla_{e_p} F(\theta') = \nabla_e F(\theta)$

- For $i \ne p$, $\nabla_{e_i} F(\theta') = 0$ (since new experts are not trained or initialized randomly and not used in forward pass)

Thus:

$$
\begin{aligned}
\|\nabla F(\theta'_1) - \nabla F(\theta'_2)\|^2 &= \sum_{i=1}^{N} \|\nabla_{e_{1,i}} F - \nabla_{e_{2,i}} F\|^2 \\
&= \|\nabla_e F(\theta_1) - \nabla_e F(\theta_2)\|^2 \\
&\le L^2 \|e_1 - e_2\|^2 \\
&\le L^2 \|\theta'_1 - \theta'_2\|^2
\end{aligned}
\tag{75}
$$

Taking square roots:

$$\|\nabla F(\theta'_1) - \nabla F(\theta'_2)\| \le L\|\theta'_1 - \theta'_2\| \tag{76}$$

$\square$

### D.5.2. SHRINKAGE DOES NOT INCREASE THE LIPSCHITZ CONSTANT

**Lemma D.18.** *Let the MoE model be:*

$$\theta = (e_1, e_2, \ldots, e_N) \in \mathbb{R}^{N \cdot d} \tag{77}$$

*After applying the **Shrinkage** operation, we obtain:*

$$\hat{e} = \sum_{r=1}^{N} a_r e_r, \quad with\ a_r \ge 0, \sum_{r=1}^{N} a_r = 1 \tag{78}$$

*Proof.* We first compute the gradient of $F$ with respect to the compressed expert:

$$\nabla_{\hat{e}} F = \sum_{r=1}^{N} a_r \nabla_{e_r} F \tag{79}$$

Then:

$$\|\nabla F(\hat{e}_1) - \nabla F(\hat{e}_2)\| = \left\|\sum_{r=1}^{N} a_r \left(\nabla_{e_{r,1}} F - \nabla_{e_{r,2}} F\right)\right\| \tag{80}$$

$$\leq \sum_{r=1}^{N} a_r \|\nabla_{e_{r,1}} F - \nabla_{e_{r,2}} F\|$$

$$\leq \sum_{r=1}^{N} a_r L \|e_{r,1} - e_{r,2}\|$$

$$= L \sum_{r=1}^{N} a_r \|e_{r,1} - e_{r,2}\|$$

Now, since:

$$\|\hat{e}_1 - \hat{e}_2\| = \left\|\sum_{r=1}^{N} a_r (e_{r,1} - e_{r,2})\right\| \leq \sum_{r=1}^{N} a_r \|e_{r,1} - e_{r,2}\| \tag{81}$$

We conclude:

$$\|\nabla F(\hat{e}_1) - \nabla F(\hat{e}_2)\| \leq L \sum_{r=1}^{N} a_r \|e_{r,1} - e_{r,2}\| \leq L \|\hat{e}_1 - \hat{e}_2\| \tag{82}$$

$$\square$$

### D.5.3. SUMMARY

- **Expansion** introduces new parameters but does **not affect** the original expert or its gradients $\Rightarrow$ **Lipschitz constant unchanged**.

- **Shrinkage** is a **convex combination** of experts $\Rightarrow$ **gradient Lipschitz constant preserved**.

- Hence, **both operations preserve the convergence properties** of federated learning algorithms under standard smoothness assumptions.

### D.6. Time Complexity Analysis

To ensure a fair comparison across all five methods, we adopt a uniform set of driving factors when deriving per-round time complexity:

- **K** – total number of clients;

- **C** – number of classes in the classification task;

- **d** – dimension of each prototype / feature vector (used for FedProto, FedTGP and FedSA);

- **P** – parameter count of the lightweight homogeneous model exchanged in FedMRL ($P \ll M$);

- **B** – local batch count per client per round;

- **E** – number of local epochs executed on each client;

- **M** – total parameter count of a client-side heterogeneous model;

- **S** – server-side training iterations required by FedTGP's contrastive step.

With these quantities we decompose the cost into three common phases: local forward-and-backward training, server aggregation (or prototype / anchor optimisation), and bidirectional communication. The resulting expressions therefore reflect the worst-case computational workload and communication volume that each algorithm imposes in a single federated round.

- **FedKD.** Each client trains its heterogeneous model and performs forward propagation on the broadcast homogeneous model to generate soft labels for distillation. Every forward/backward pass involves the full model parameters $M$, repeated for $E$ epochs and $B$ batches per epoch, giving a local compute cost of $E \cdot B \cdot M$. With $K$ clients running in parallel, the total client-side cost is $K \cdot E \cdot B \cdot M$. The server only averages the uploaded homogeneous models, yielding $O(KM)$ computation and communication, which is absorbed by the client term. Hence the per-round complexity is $O(K \cdot E \cdot B \cdot M)$.

- **FedProto.** Local training is identical to FedKD, hence $K \cdot E \cdot B \cdot M$ is retained. After training, each client computes $C$ class prototypes by averaging $d$-dimensional features; the cost $C \cdot d$ is negligible compared with $M$. The server collects and averages these prototypes from $K$ clients, requiring $O(K \cdot C \cdot d)$ computation and communication. The overall per-round complexity becomes $O(K \cdot E \cdot B \cdot M + K \cdot C \cdot d)$.

- **FedMRL.** Every client runs both its heterogeneous model and the global homogeneous small model once per sample, concatenates their representations, and feeds the result into a lightweight projector. The projector's parameter count is $\ll M$, so the local cost is still dominated by $E \cdot B \cdot M$. With $K$ clients the client-side total is $K \cdot E \cdot B \cdot M$. The server aggregates only the homogeneous small model whose size is denoted by $P \ll M$, giving $O(K \cdot P)$ computation and communication. Thus the per-round complexity is $O(K \cdot E \cdot B \cdot M + K \cdot P)$.

- **FedTGP.** Local training and prototype extraction are the same as in FedProto, contributing $K \cdot E \cdot B \cdot M + K \cdot C \cdot d$. Instead of simple averaging, the server performs $S$ iterations of contrastive learning among $C$ trainable prototypes. Each iteration computes $O(C^2)$ pairwise distances of dimension $d$, adding $S \cdot C^2 \cdot d$ server-side computation. Communication remains prototype-based, i.e., $O(K \cdot C \cdot d)$. The total per-round complexity is therefore $O(K \cdot E \cdot B \cdot M + K \cdot C \cdot d + S \cdot C^2 \cdot d)$.

- **FedSA.** Clients conduct standard model training ($E \cdot B \cdot M$) and then align features with $C$ semantic anchors of dimension $d$; the anchor-related cost $C \cdot d$ is negligible versus $M$. With $K$ clients the client-side total is $K \cdot E \cdot B \cdot M$. The server only averages the anchor vectors, yielding $O(K \cdot C \cdot d)$ computation and communication. Hence the per-round complexity is $O(K \cdot E \cdot B \cdot M + K \cdot C \cdot d)$.

- **FedEMoE.** Every client runs E epochs of B mini-batches over its heterogeneous MoE , giving a client compute cost of $K \cdot E \cdot B \cdot M$. After training, each client uploads only the sparse SE sub-experters it touched together with the full FE; these are subsets of $M$, so upstream traffic is $K \cdot M$. The server first does the usual weighted/sparse aggregation (cost absorbed in the upload). It then refines each of the $N$ SE sub-experts (width $d$) by one SVD, adding $Nd^3$ server-side computation. Finally the refined global model is broadcast back, contributing another $K \cdot M$ downstream. The total per-round complexity is therefore $O(K \cdot E \cdot B \cdot M + N \cdot d^3)$. From the above analysis, it can be concluded that the worst time complexity of FedEMoE is less than FedTGP but greater than other methods.

## D.7. Related Technique

### D.7.1. MoE ADJUSTMENT

*Table 9.* Comparison of dynamic-capacity MoE mechanisms. *Comp.* = additional compute cost per round. *Scene* = centralised vs federated. *Data* = extra public/validation data required. *Feat.* = whether the operation breaks the existing feature space and demands full-model re-warming.

| Method | Comp. | Scene | Data | Feat. |
|---|---|---|---|---|
| BASE Layer | Low | Centralized | Validation set | Yes (whole layer re-init) |
| Switch-Transformer | Medium | Centralized | None | Yes (new FFN weights) |
| ST-MoE | High | Centralized | Gradient buffer | Yes (expert re-randomize) |
| **FedEMoE (ours)** | **Low** | **Federated** | **None** | **No (sub-expert grafting)** |

The fundamental difference lies in feature space continuity. As shown in Table 9, existing dynamic MoE methods (BASE (Lewis et al., 2021), Switch-Transformer (Fedus et al., 2022), ST-MoE (Zoph et al., 2022)) rely on destructive capacity adjustments—re-initializing, replacing, or re-randomizing weights. This effectively "resets" the expert's learning, necessitating heavy re-training, large gradient buffers, or global warm-up cycles to recover.

In contrast, FedEMoE employs preservative elasticity. By performing lightweight SVD on the SE weight matrix locally, we decompose and restructure experts without destroying the learned feature manifold. This allows us to adjust capacity while

keeping PEG weights intact, eliminating the need for global re-warming or uploading extra data—a critical requirement for federated learning contexts.

### D.7.2. DIAGNOSTIC TECHNIQUE

*Table 10.* Comparison of expert-state diagnostic tools.

| Method | Needs Extra Data | FL-Friendly |
|---|---|---|
| Validation-loss trigger | held-out set | medium |
| Gradient-norm / Fisher | batch gradients | poor |
| Fixed schedule | none | medium |
| **Weight-spectrum (ours)** | **none** | **best** |

Weight-spectrum analysis is the only diagnostic that requires no extra data or labels, uploads no additional tensors, and provides a provable upper bound on client drift. These properties make it the most communication-efficient, privacy-preserving, and theoretically grounded choice for elastic capacity control in federated learning.

## E. Implementation Details

### E.1. Hyperparameters settings

The key hyperparameters used in our experiments are summarized in Table 11.

*Table 11.* Summary of key hyperparameters

| Hyperparameter | Value |
|---|---|
| Framework | PyTorch 2.1.0 (CUDA 12.1, NVIDIA GPUs) |
| Optimizer | Adam (for client updates) |
| Local learning rate ($\eta$) | 0.005 |
| Local epochs | 1 |
| Batch size (local training) | 64 |
| Data size for knowledge exchange | 64 |
| KD temperature (**T**) | 2.0 |
| Number of shared MoE experts ($N$) | 3 |
| Shared MoE top$K$ | 2 |
| Communication rounds ($T$) | 100 |
| Client participation rate | 0.2 |
| Data heterogeneity ($\beta$ for Dirichlet) | 0.01, 0.1, 0.5, 10 |
| Scaling operation gap ($\mathcal{T}$) | 20 |
| Initial expansion threshold ($v_0$) | 0.8 |
| EMA smoothing factor ($\gamma$) | 0.07 |
| Energy ratio ($\theta$) | 0.99 |
| Fraction of top components to drop ($\alpha$) | 0.9 |
| Tail energy proportion threshold | 0.05 |
| Random seed | 1 |

### E.2. Model Settings

In our main experiments, the models are based on the MoE architecture. This choice leads to model heterogeneity (Table 12) that comes from several factors. The main ones are the differences in model structure caused by varying the number of experts in the MoE setup (Table 13) and the number of experts activated during processing. Then there's the personalized composition of expert groups, which varies across models. Each model has its own combination of expert types. Also, the internal structures of the experts in Table 14 and classifiers themselves can differ, affecting how they process features and make decisions. In terms of model size, the models we set up in our experiments are similar to ResNet18 (11.2MB). For

other backbone networks (ResNet, MobileNet, GoogleNet), we replace the second fully connected layer in their network with the MoE layer in the Table 13.

*Table 12.* CNN Configurations used in Heterogeneous Experiments.

| Name | Component | Params |
|------|-----------|--------|
| $CNN_1$ | 1 Conv layer, $MoELayer_1$, 1 FC Layer | 10.44M |
| $CNN_2$ | 1 Conv layer, $MoELayer_2$, 1 FC Layer | 11.37M |
| $CNN_3$ | 1 Conv layer, $MoELayer_3$, 1 FC Layer | 14.35M |
| $CNN_4$ | 1 Conv layer, $MoELayer_4$, 1 FC Layer | 12.08M |
| $CNN_5$ | 1 Conv layer, $MoELayer_5$, 1 FC Layer | 7.06M |

*Table 13.* MoE Configurations used in Heterogeneous CNNs.

| Name | Component | (N,topK) | Params |
|------|-----------|----------|--------|
| $MoELayer_1$ | 3 $Expert_1$, 1 $Expert_2$,1 $SE(e^*)$ ,1 FC Layer | (5,4) | 10.31M |
| $MoELayer_2$ | 2 $Expert_1$, 1 $Expert_2$,1 $Expert_3$, 1 $SE(e^*)$, 1 FC Layer | (5,3) | 11.25M |
| $MoELayer_3$ | 1 $Expert_1$, 1 $Expert_2$,3 $Expert_3$, 1 $SE(e^*)$, 1 FC Layer | (6,4) | 14.21M |
| $MoELayer_4$ | 2 $Expert_2$, 2 $Expert_3$, 1 $SE(e^*)$, 1 FC Layer | (5,3) | 11.95M |
| $MoELayer_5$ | 2 $Expert_2$, 1 $SE(e^*)$, 1 FC Layer | (2,1) | 7.01M |
| $SE(e^*)$ | 3 $Expert_4$, 1 FC Layer | (3,2) | 0.05M |

*Table 14.* Expert Configurations used in Heterogeneous MoE layer.

| Name | Component | Params |
|------|-----------|--------|
| $Expert_1$ | 2 Conv layers, 1 FC layer | 0.85M |
| $Expert_2$ | 1 Conv layers, 1 FC layer | 1.07M |
| $Expert_3$ | 1 FC layer1 | 2.01M |
| $Expert_4$ | 1 FC layer1 | 0.02M |

# F. Additional Results

### F.1. Main Experiments

Experimental results in Table 15 on CIFAR-10, CIFAR-100, and Tiny-ImageNet—under both heterogeneous and homogeneous model settings with $K = 20$ demonstrate that the proposed method achieves the highest test accuracy and significantly outperforms existing federated learning algorithms. Specifically, on CIFAR-10, it surpasses the runner-up by 3.21% (heterogeneous) and 2.01% (homogeneous); on the more challenging CIFAR-100, the gains reach 21.07% and 24.41%; and on Tiny-ImageNet, it leads by 11.18% and 3.33%, respectively. Collectively, these results underscore the superior generalization capability of our approach.

In the setting of $K = 100$ clients, we evaluate the test accuracies of all methods on CIFAR-10, CIFAR-100, and Tiny-ImageNet under both heterogeneous and homogeneous model scenarios. As shown in Table 16, our method attains 78.26% and 78.84% in the heterogeneous and homogeneous cases, respectively, outperforming the runner-up FedGC by 0.38% and 0.74%. On CIFAR-100, it achieves 41.57% and 41.76%, surpassing FedGC by 16.54% and FedTGP by 18.67%. On Tiny-ImageNet, our method reaches 25.35% in the heterogeneous scenario, a 10.84% gain over FedAvg (22.87%), and 25.82% in the homogeneous scenario, an improvement of 48.05% over FedSA (17.43%).

### F.2. Convergence Speed Analysis

On CIFAR-10 with 20 clients, the accuracy curves of FedKD, FedTGP, FedProto, FedSA and FedMRL drift around 60% as the communication rounds advance from 0 to 80. In contrast, our approach (Ours) surpasses 80% within only 20 rounds and stays flat thereafter, opening a clear margin of 10–20%. When the client scale grows to 50, all baselines slow down further and still plateau near 70%. However, our approach breaks the 80% barrier around round 20 and finishes more than 15% ahead, demonstrating excellent scalability. In the extremely sparse 100-client setup, the rival curves oscillate violently and

*Table 15.* Test accuracy(%) of client with K = 20 under different datasets in model-heterogeneous and model-homogeneous scenarios.

| Algorithm | CIFAR-10 | CIFAR-100 | Tiny-Imagenet |
|---|---|---|---|
| FedAvg | 78.79 | 38.54 | 22.74 |
| FedGC | 80.57 | 40.33 | 25.83 |
| FedALA | 79.64 | 35.06 | 15.86 |
| FedPAC | 78.62 | 33.42 | 20.61 |
| FedDBE | 79.12 | 38.65 | 22.85 |
| Ours | 83.16(3.21%) | 48.83(21.07%) | 28.72(11.18%) |
| FedKD | 76.81 | 33.62 | 18.24 |
| FedProto | 70.74 | 29.12 | 6.93 |
| FedMRL | 81.84 | 39.19 | 18.74 |
| FedTGP | 68.42 | 35.64 | 12.61 |
| FedSA | 76.01 | 39.41 | 28.18 |
| Ours | 83.48(2.01%) | 49.03(24.41%) | 29.12(3.33%) |

*Table 16.* Test accuracy(%) of client with K = 100 under different datasets in model-heterogeneous and model-homogeneous scenarios.

| Algorithm | CIFAR-10 | CIFAR-100 | Tiny-Imagenet |
|---|---|---|---|
| FedAvg | 75.42 | 31.36 | 22.87 |
| FedGC | 77.96 | 35.67 | 21.65 |
| FedALA | 76.23 | 31.41 | 20.16 |
| FedPAC | 75.93 | 16.24 | 17.63 |
| FedDBE | 76.07 | 31.16 | 18.81 |
| Ours | 78.26(0.38%) | 41.57(16.54%) | 25.35(10.84%) |
| FedKD | 71.57 | 29.41 | 17.44 |
| FedProto | 68.38 | 23.11 | 1.51 |
| FedMRL | 78.26 | 35.52 | 17.21 |
| FedTGP | 68.13 | 35.19 | 11.33 |
| FedSA | 57.48 | 31.65 | 17.43 |
| Ours | 78.84(0.74%) | 41.76(18.67%) | 25.82(48.05%) |

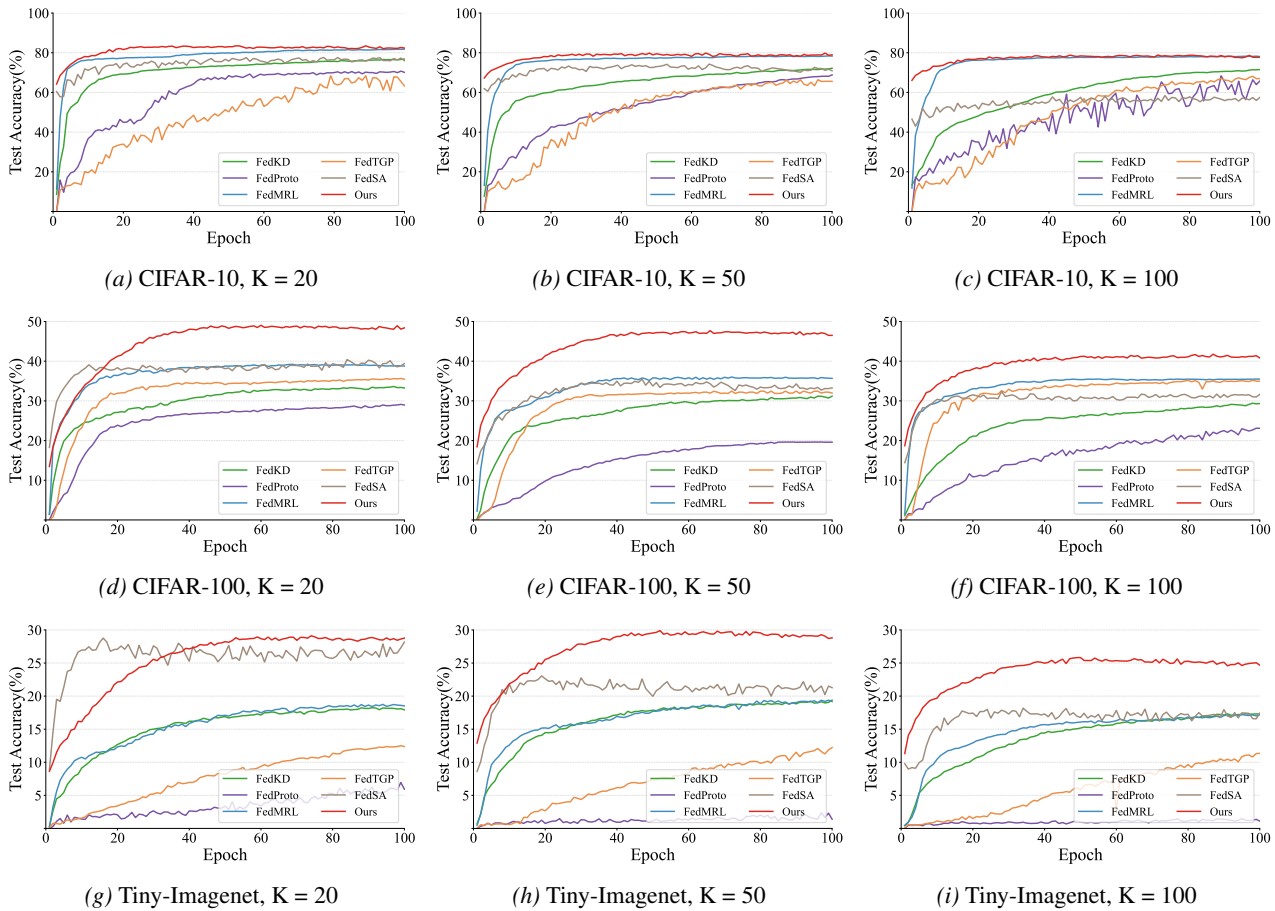

*Figure 7.* Comparison of Federated Learning Approaches' Performance Across Datasets and Client Scales.

FedKD, FedProto, etc. even drop below 60%. Ours remains stable, climbing to 80% around round 20 and maintaining a final gap of roughly 20%, proving its robustness under high decentralization.

Switching to the harder CIFAR-100 task, top-1 accuracies plunge below 50% for 20 clients; FedMRL and FedSA linger around 40%. Ours is the first to exceed 50% within the same schedule and ends up about 10% higher, revealing stronger discrimination for fine-grained classes. At 50 clients on CIFAR-100, the baselines stay in the 30%–40% band while Ours crosses 50% shortly after round 20 and keeps a 15% lead at the end, validating superior generalization when both class count and data heterogeneity increase. Even in the extreme 100-client CIFAR-100 scenario, Ours still reaches above 45% around round 20 whereas FedKD, FedProto, etc. remain near 30%, yielding a terminal advantage of about 15% and highlighting highly efficient knowledge aggregation under severe fragmentation.

Moving to the larger-scale Tiny-ImageNet experiments, none of the baselines exceed 25% with 20 clients; Ours breaks 30% around round 20 and finishes roughly 5% ahead. When clients increase to 50 and 100, the competitors never surpass the 25% ceiling, while Ours stays consistently above 30% and holds a stable 5–7% margin, confirming steady gains on high-resolution, thousand-class data.

Across all tested conditions, Ours consistently delivers faster convergence, higher final accuracy and smoother curves, comprehensively outperforming state-of-the-art alternatives and offering a reliable solution for real-world cross-device deployment.

### F.3. Decoupling Ability Experiment

We compare FedEMoE with other decoupling-based PFL methods(DualFed (Zhu et al., 2024), GPFL (Zhang et al., 2023b), FedDecomp (Wu et al., 2024), FedCAC (Wu et al., 2023)) that also separate personal and shared components. Table 17

reports CIFAR-100 accuracies with homogeneous CNNs (K=50, Dir(0.1)). FedEMoE outperforms the strongest competitor FedCAC by 0.54% and beating DualFed by 4.15%, which shows that our spectrum-driven elastic SE extracts richer general knowledge than the fixed-capacity shared branches used in prior methods.

*Table 17.* Comparison with decoupling-based PFL methods.

| Method | DualFed | GPFL | FedDecomp | FedCAC | **Ours** |
|---|---|---|---|---|---|
| Accuracy (%) | 42.32 | 31.74 | 36.79 | 45.93 | **46.47** |

## F.4. Additional Sensitivity Analysis

Table 18 explores the effect of the knowledge-distillation temperature $\mathcal{T}$. We keep the rest of the model fixed at three shared expert $N = 3$ and $topK = 2$, and we vary $\mathcal{T}$ from 1 to 4. The highest accuracy (49.03%) is reached when $\mathcal{T} = 2$. A lower temperature produces an overly sharp probability vector that limits guidance, whereas a higher one flattens the teacher signal and slows convergence. The 0.3% spread across the whole range shows that the method is not sensitive to the exact value.

*Table 18.* Sensitivity to KD temperature $\mathcal{T}$ (3 experts, topK=2).

| $\mathcal{T}$ | 1 | 2 | 3 | 4 |
|---|---|---|---|---|
| Accuracy (%) | 48.71 | **49.03** | 48.98 | 48.96 |

## F.5. Measurement of MoE-Induced Computational Overhead

To quantitatively evaluate the extra computational burden from the MoE architecture, we compared the model without MoE against the full FedEMoE model under identical conditions. The model without MoE requires **0.961 GFLOPs** during training (including both forward and backward propagation), whereas the full FedEMoE model averages approximately **1.34 GFLOPs**. The difference of about **0.38 GFLOPs** is primarily attributable to the lightweight components of the MoE architecture, such as the sparse gating network and small-scale expert modules. The computational overhead introduced by the MoE architecture is only a small increase and is fully acceptable given the substantial gains in personalization accuracy.

## F.6. Impact of Expert Status Diagnosis on Performance Stability

We designed an experiment to verify model performance change and the convergence of the number of experts. Table 19 reports the maximum accuracy and the number of experts at each training stage. In the early stages (T=0-19), underfitting experts trigger expansion to increase model capacity, causing the expert count to grow from 3 to 9. In the mid-to-late stages (T=40-99), expert states gradually stabilize, and the number of experts converges to a relatively stable value of 15. The accuracy improves monotonically from 31.72% to 49.03% and then slightly stabilizes at 48.97%, demonstrating that our elastic mechanism achieves convergence without performance degradation.

*Table 19.* Periodic performance (T=20) variation of FedEMoE on CIFAR-100 with 20 clients.

| T | 0-19 | 20-39 | 40-59 | 60-79 | 80-99 |
|---|---|---|---|---|---|
| **Max Acc (%)** | 31.72 | 42.81 | 47.66 | 49.03 | 48.97 |
| **Number of Experts** | 3 | 9 | 13 | 15 | 15 |

