# OpenReview forum: "FedEMoE: Improving Personalization on Heterogeneous Federated Learning via Elastic Mixture of Experts Architecture"
_ICML.cc/2026/Conference — ICML 2026 regular_

### Official Review · Reviewer_ASnD · 2026-03-09

**Soundness:** 3
**Presentation:** 3
**Significance:** 3
**Originality:** 3
**Overall Recommendation:** 4
**Confidence:** 3

**Summary:**

This paper addresses the challenge of intertwined personalized and generalized knowledge in Heterogeneous Federated Learning (HtFL), which often hinders effective knowledge decoupling. To resolve this, the authors propose a novel framework named FedEMoE. The core contribution lies in the introduction of an Elastic Mixture of Experts (EMoE) architecture, which decouples the model structure into a Personalized Expert Group (PEG) dedicated to capturing local characteristics and a Shared Expert pool (SE) responsible for extracting global common knowledge.
The key innovations of FedEMoE include: (1) Multi-scale Feature Extraction: Utilizing PEG to enrich local discriminative information; (2) Spectrum-driven Elastic Selection: Dynamically adjusting the capacity of shared experts through weight spectrum analysis to adapt to the resource constraints of various clients; (3) Bidirectional Knowledge Distillation: Facilitating the synergistic evolution of knowledge between PEG and SE during local training. Experimental results demonstrate that FedEMoE excels in handling model heterogeneity and non-IID data issues, significantly improving personalization performance in heterogeneous environments.

**Compliance With Llm Reviewing Policy:**

Affirmed.

**Key Questions For Authors:**

Q1. Analysis of Expert Utilization and Load Balancing: In the proposed Top-K gating mechanism, does a rich-get-richer effect occur where a small subset of shared experts (SE) dominates the weights while others are rarely activated? Given the significant data heterogeneity (non-IID) across clients, it is unclear whether the framework can effectively prevent expert collapse and ensure that all experts in the pool are sufficiently and uniformly trained during the collaborative process.
Q2. Semantic Alignment in Heterogeneous Backbone Scenarios: The framework claims to support heterogeneous model architectures. However, when different clients use backbones with fundamentally different operations, the feature maps generated by the Personalized Expert Group (PEG) may have entirely different semantic structures and statistical distributions. How does FedEMoE ensure that the subsequent Top-K gating and shared expert layers can correctly interpret and aggregate these semantically misaligned features without a dedicated projection or alignment layer?
Q3. Viability of Spectrum-driven Strategy on Resource-constrained Devices: The spectrum-driven elastic strategy relies on performing singular value decomposition (SVD) or similar spectral analysis on weight matrices during local training to determine the effective rank. This introduces a non-trivial computational burden that seems to contradict the very goal of serving resource-constrained edge devices.

**Limitations:**

yes

**Strengths And Weaknesses:**

Strengths
1. Innovative Elastic Decoupled Architecture: The paper ingeniously introduces the Mixture of Experts (MoE) architecture into the field of heterogeneous federated learning. By categorizing experts into "personalized" and "shared" types and employing a gating network for sparse activation, it fundamentally addresses the interference between personalized and generalized knowledge in PFL.
2. Spectrum-driven Resource Adaptation Strategy: The introduction of weight spectrum analysis to dynamically adjust the Effective Rank of shared experts is a technically rigorous and well-justified design. This enables the model to adjust expert capacity based on training states and hardware constraints, providing a solid mathematical foundation for load balancing across heterogeneous devices.
3. Extensive Experimental Validation: The experimental section is exceptionally solid. The authors compare their method against over 10 state-of-the-art (SOTA) methods (e.g., FedRolex, FedPAC, FedTGP, etc.) across various datasets (CIFAR-10/100, Tiny-ImageNet) and heterogeneous scenarios. The inclusion of convergence proofs and complexity analysis in the appendix further strengthens the academic depth of the paper.
4. Superior Synergistic Component Design: The design of multi-scale feature concatenation and double knowledge distillation (Double KD) is highly logical. The former ensures feature discriminability, while the latter enables efficient exchange between global and local knowledge, creating a cohesive and closed-loop logic for the entire framework.
Weaknesses
1. Gating Network Stability under Extreme Heterogeneity: Although Top-K gating enables sparse activation, the gating network may suffer from "Expert Collapse"—where a few experts are over-activated while others are neglected—in extreme non-IID scenarios (e.g., Dirichlet $\alpha=0.01$). The paper provides relatively thin argumentation regarding the load-balancing mechanism of the gating network.
2. Communication Risks Associated with Increasing Experts: As the number of experts $N$ increases, even though only a sparse subset is activated per round, the server must still maintain the weights of the entire expert pool. The paper lacks stress testing for server storage pressure and the transmission overhead of gating parameters under large-scale expert configurations.
3. Dependency on Local Data Volume: Due to the multi-expert architecture, the total number of model parameters (especially in the PEG) increases significantly. For "cold-start" clients with very few local samples, there is a potential risk of severe overfitting within this complex architecture, a concern not fully discussed in the experimental section.

---

> ### Author Rebuttal · Authors · 2026-03-30
>
> We thank the reviewer for taking the time to review our work and for providing thoughtful feedback. In the following, we provide detailed responses to each concern.
>
> **R1: FedEMoE can address the expert collapse risk.**
> **Our core idea of FedEMoE is: we expect that each expert specialize in specific data distributions, not to aim traditional load balancing.** Unbalanced activation precisely reflects that the model has formed an effective division of labor cooperation. Specifically,  each expert can be activated by similar samples in the uncontrolled gating mechanism, focusing on itself knowledge.
>
> The key way for preventing expert collapse lies in the elastic mechanism: when a group of experts remains in a low-activation state, they are merged through shrinkage operation; on the contract, when an expert is over-activated, it is split into multiple sub-experts through expansion operation.
>
> **R2: FedEMoE has taken into account the issue of semantic alignment.** Our model architecture is FE - Backbone - PEG \& SE - Head, where the Personalized Expert Group (PEG) and the Shared Expert (SE) are under the same Top-K routing mechanism (SE and PEG are parallel modules). In this design, the PEG is used exclusively for local personalized feature extraction and does not affect the input distribution of the SE. We design a bidirectional Knowledge distillation between PEG and SE to complete knowledge exchange. Consequently, the PEG does not produce semantic alignment issues.
>
> As detailed in the Appendix E.2, We introduce a projection layer at the end of each client’s private backbone. The parameters of this projection layer can map the heterogeneous features output by the heterogeneous backbone into a unified semantic space. We add this explanation in Section 4 in our manuscript.
>
> **R3:** Thanks for your good comments. **The spectrum-driven strategy in FedEMoE is effective and suitable.** The specific reasons are as follows:
>
>    - Our spectrum-driven strategy is **performed on the server side**, where resources are considered sufficient or unlimited.
>    - **SVD diagnosis is executed periodically at a default interval** (T=20 in our experiments), accounting for a very small proportion of the total computational cost.
>     - SVD is only applied to the weight matrix of each expert within the SE, where **the parameter size of each expert is small (only 4096 parameters)**, consuming 0.0031 GFLOPs.
>   - **The current diagnosis of SVD is accurate and efficient as shown in Table 9 of Appendix D.7.2**, compared to other diagnosis methods.
>
> In a nutshell, **SVD spectrum diagnosis achieves a balance between resource consumption and diagnostic effectiveness.**

---

### Official Review · Reviewer_RAvX · 2026-03-11

**Soundness:** 2
**Presentation:** 2
**Significance:** 3
**Originality:** 3
**Overall Recommendation:** 4
**Confidence:** 3

**Summary:**

This paper introduces FedEMoE, an evolving MoE paradigm that can adjust the model capacity during training. Numerical results show performance gain of the proposed method compare to baselines.

**Compliance With Llm Reviewing Policy:**

Affirmed.

**Final Justification:**

The authors addressed my concerns and I increase my rating accordingly.

**Key Questions For Authors:**

See weakness.

**Strengths And Weaknesses:**

Pros:
1. Well-designed method and well-illustrated motivation. The designed evolving MoE method is novel and may have high impact.
2. The authors provide comprehensive experiments and ablation studies.

Cons:
1. I'm confusing about the sparse model aggregation. From Eq 5, the whole sub-experts are updated based on the activated clients. However, in Figure 2, part of weights are updated.

2. The effectiveness of Expert Status Diagnosis should be discussed. For example, will the model performance change significantly when adding/removing experts? Will the number of experts converge? Please also provide the activation analysis and the load-banlance analysis.

3. It's not sure whether the performance evaluation is done on global test dataset or evaluating personalization performance. If is later, the comparision may not fair and lack comparision to personalized FL methods. Besides, do the baselines use the same model size as FedEMoE?

4. Please report the training/inference wall-clock time. Besides, how the expert parallel implemented for such a dynamic MoE apporach?

---

> ### Author Rebuttal · Authors · 2026-03-30
>
> We thank the reviewer for their valuable insights and suggestions. Below we provide point-by-point responses to each concern.
>
> **R1:** To avoid ambiguity, we clarify that Equation (5) describes the server-side aggregation for **each sub-expert that is activated and uploaded by all participated clients**. However, Figure 2 is intended to illustrate the differences in aggregation methods. The complete sparse aggregation logic is as follows:
>
> - Clients only upload activated sub-experts (a subset of the overall sub-experts in the SE).
> - FedEMoE aggregates grouped sub-experts on server side.
>
> We have revised the description of Equation (5) to explicitly state the premise of "based on the set of clients that activated this sub-expert" and revised the caption of Figure 2 to "The visual case of difference between dense and sparse aggregation" for eliminating ambiguity.
>
> **R2:** We thank the reviewer for the insightful questions. While blindly altering the number of experts heavily impacts model capacity and could potentially lead to unstable performance, our Expert Status Diagnosis is highly effective because it acts as a precise, data-free guider. By employing weight spectrum analysis, it accurately identifies whether an expert is under-fitting or over-fitting. Guided by this accurate diagnosis, our elastic mechanism transforms potential fluctuations into stable, monotonic improvements.
>
> As shown in Table II, we report the maximum accuracy and the number of experts at each stage under the experimental setup on CIFAR-100 with 20 clients. In the early stages, underfitting experts trigger expansion to increase model capacity; in the mid-to-late stages, expert states stabilize, and **the number of experts converges to a relatively stable value**. We have added the below Table II in the paper to further explain this convergence process.
>
> Moreover, **our core idea of FedEMoE is: we expect that each expert specialize in specific data distributions, not to aim traditional load balancing.** Based on this design, FedEMoE can adaptively modify model structures by the elastic mechanism, enabling models to better fit the current tasks and data distributions. Meanwhile, our amount of sub-expert is changed, we are hardly to count the activation times of sub-experts.
>
> **Table II:** Periodic performance (T=20) variation of FedEMoE on CIFAR-100 with 20 clients.
>
> | T           | 0-19 | 20-39 | 40-59 | 60-79 | 80-99 |
> |-------------|------|-------|-------|-------|-------|
> | **Max Acc(%)** | 31.72 | 42.81 | 47.66 | 49.03 | 48.97 |
> | **Number**     | 3     | 9     | 13    | 15    | 15    |
>
> **R3:** In our experiments, following standard pFL framework[r1-r2], the personalization performance evaluation of all methods in our experiments is based on the local test set. Crucially, our baselines extensively include SOTA pFL methods, such as FedALA, FedPAC, and FedDBE for homogeneous settings, and FedProto, FedMRL, and FedTGP for heterogeneous settings. Moreover, across the model settings for all methods are kept the same initial model as FedEMoE. Therefore, the comparison is highly fair for pFL.
>
> [r1] Pfllib: A beginner-friendly and comprehensive personalized federated learning library and benchmark. JMLR, 26(50):1–10, 2025.
>
> [r2] Htfllib: A comprehensive heterogeneous federated learning library and benchmark. In Proceedings of the 31st ACM SIGKDD 2025.
>
> **R4:** Referring to Table I in the response to the Reviewer Tbta, compared with the baselines, FedEMoE achieves better computational efficiency. In addition, the training and inference time per round on a single client of FedEMoE (3.6s) is less than that of FedMRL (4.1s).
>
> Our MoE structure on SE adopts a nested model architecture (e.g., $[e_{1,1}, [e_{2,1}, e_{2,2}, e_{2,3}], e_{1,3}]$).
> Similar to other MoE approaches, experts within the same hierarchical level (e.g., $e_{1,1}$ and $e_{1,3}$) are processed in parallel, whereas experts across different hierarchical levels (e.g., $e_{1,1}$ and $e_{2,1}$) are processed sequentially.

---

> > ### Author Rebuttal · Reviewer_RAvX · 2026-04-02
> >
> > The authors addressed my concerns and I increase my rating accordingly.

---

> > > ### Author Response · Authors · 2026-04-05
> > >
> > > Sincere thanks for raising your score and for your recognition. We have revised paper accordingly based on your comments.

---

### Official Review · Reviewer_cZvF · 2026-03-11

**Soundness:** 2
**Presentation:** 2
**Significance:** 3
**Originality:** 3
**Overall Recommendation:** 4
**Confidence:** 3

**Summary:**

This paper introduces the FedEMOE system, a novel architecture that decouples personalization from generalization using an Elastic Mixture of Experts (EMOE). The aim is to be able to train both generalizable and personalized models while minimizing communication costs. It first uses a multi-scale feature extraction mechanism to enrich local, personalized knowledge. Then they use an elastic shared expert model which overcomes knowledge transfer bottlenecks between heterogeneous clients by adaptively expanding or shrinking based on weight spectrum analysis. They apply sparse model aggregation to reduce communication cost, and uses knowledge distillation to better train the local personalized models better. Experiments are conducted on Cifar and TinyImagenet datasets across various scales and heterogeneity systems. Comparison against state of the art shows that they outperform them comfortably across various dimensions.

**Compliance With Llm Reviewing Policy:**

Affirmed.

**Key Questions For Authors:**

1. Please elaborate on what is meant by "capacity" in "Its capacity can adaptively expand or shrink based on collective training". Is it number of experts? Size of features retained by experts? How so?
2. How does this scaling help with cost?
3. Please explain more about the reasoning behind why a multi-scale feature extractor is needed?
4. How many experts in the MoE for the setups?
5. Highly important information, especially in regards to the story are delegated to the Appendix section for the Expert Evolution section. It is suggested to include them, at least in summary, and if space is a constraint then Figures 2 and 3 can be move to appendix since. Otherwise following the reasoning becomes challenging.
6. Multiple claims on reduction of memory, compute, bandwidth etc. are made, but are not backed up by results. It seems that it will reasonably do so, but how much exactly, especially compared to the other methods is still under question.
7. Multiple grammatical/spelling mistakes (e.g. none which, statusment, actived, erpert, etc.)

**Limitations:**

Yes.

**Strengths And Weaknesses:**

Strengths -
1. Interesting, albeit complicated, solution and approach.
2. Results are very good across the important aspects, especially compared to the state of the art.

Weaknesses -
1. Seems much more expensive computationally. More details needed on cost.
2. A lot of the writing on the implementation details and reasoning is unclear.

---

> ### Author Rebuttal · Authors · 2026-03-30
>
> We appreciate the reviewer’s careful review and constructive comments. We have addressed all the points below and hope this clarifies our work.
>
> **R1:** In this paper, **“capacity” specifically refers to how much data  can be adapt to on experts. This capacity adjustment is achieved by careful modification of experts number.** When an expert is in an underfitting state, we increase the number of its sub-experts through expansion, thereby enhancing its fitting capability; when some experts are in overfitting states, we compress redundant parameters through shrinkage. In addition, the effective rank in weight spectrum analysis directly reflects the actual utilized dimensions of the parameter matrix in the feature space, making it a core indicator for capacity diagnosis.
>
> **R2:** The elastic mechanism contributes to computational cost savings in three ways:
>
> - **The shrinkage operation directly reduces the number of model parameters by merging redundant expert parameters**, thereby lowering the computational overhead associated with local storage, communication transmission, and subsequent training.
>
> - **The expansion operation** preserves the original expert’s weights, allowing the model to gain additional capacity without retraining from scratch, **effective avoiding the global warm-up or extensive additional training rounds as existing methods usually do.**
> - We further design the dynamic threshold adjustment mechanism with Exponential Moving Average (EMA), which can limit invalid sub-experts increasing.
>
> **R3:** The design of the Personalized Expert Group (PEG) is motivated by **the heterogeneity of data distributions and the requirement for local personalization on each client.** In real world scenarios, the data distributions across different clients are heterogeneity. Moreover, it is hard to simultaneously adapt to samples from different classes for a single fixed-scale feature extractor. To address this, PEG employs multiple parallel feature extraction networks to capture feature representations at different scales. Furthermore, the gating mechanism adaptively selects the optimal feature combination for each input sample, allowing individual clients to selectively utilize information most pertinent to their respective data distributions.
>
>
> **R4:** In our experimental setup, the Mixture-of-Experts (MoE) of the Shared Expert (SE) initially contains 3 experts, each of which is a simple neural network. For the Personalized Expert Group (PEG) component, the number of personalized experts varies across different client models. The detailed configurations can be found in Appendix E.2.
>
> **R5:** Thanks for your wonderful suggestion. We add the summary and move Figures 2 and 3 to Appendix section. The Expert Evolution Mechanism is one of the core contributions of FedEMoE. And this related content in the appendix does affect the readability of the main text.
> We have moved the Figure 2 to appendix and added the core content to the main text.
>
> **R6:** We acknowledge that the description of resource conservation in the paper lacks quantitative comparative data.
>  - **FedEMoE uploads fewer parameters.** Compared with other model-based methods, FedEMoE reduces communication cost by uploading only the lightweight and sparse SE instead of the entire model.
>  - **Memory and storage consumption are lowered.** The shrink operation reduces the number of experts, directly decreasing the storage size and computational consumption required by the model.
> - **Computational cost remains efficient in FLOPs.** The extra floating-point operations of FedEMoE amount to 1.34 GFLOPs, which is superior to other model-based federated learning methods.
> - **Computation time per round is competitive.** Under the same experimental setting, FedEMoE achieves a computational time of 93 seconds per round, which is superior to most baselines, except for FedMRL (91 seconds per round). A detailed comparison of resource consumption is presented in Table I in the response to Reviewer Tbta.
>
> **R7:** Thanks for your valuable comments. We fixed all the mentioned above typos, grammar errors and double checked others.

---

> > ### Author Rebuttal · Reviewer_cZvF · 2026-04-02
> >
> > The resource overhead concern was not directly addressed, but the others were. From the other reviewer responses, it is clear that there is extra cost on compute which may be a hurdle for widespread adoption. As such, I keep my score, unless the authors can justify it.

---

> > > ### Author Response · Authors · 2026-04-02
> > >
> > > In federated learning, personalization or knowledge transfer mechanisms always come with some overhead. Our FedEMoE is no exception, but as shown below, its overhead is acceptable and highly rewarding.
> > >
> > > 1. **The extra computation cost is acceptable.** On a device with poor computing power (1 GFLOP/s), this costs only 0.8 seconds more than FedProto per round. However, our training stability is better than FedProto and FedTGP.
> > >
> > > 2. **The computational overhead of our method is manageable and can be readily addressed in practice.** In FedEMoE, the main cost comes from the singular value decomposition (SVD) on the server-side and the server computation is far less constrained in real-world deployment.
> > >
> > > 3. **The modest overhead delivers significant performance gains.** FedEMoE achieves up to +48.05% accuracy on Tiny-ImageNet (Table 3) and +32.43% on CIFAR-100 (Table 1). From a cost-benefit perspective, an extra ~1.24 GFLOPs per round is **a small price** for such gains in performance.
> > >
> > > 4. **Our method achieves the best performance with minimal extra computational cost.** To further evaluate this, we measure the resources required for each method to reach 35% accuracy. As shown in Table I, FedEMoE reaches this target in only 17 rounds (vs. 38 for FedMRL, >100 for others) and consumes only 21.1 GFLOPs — less than half of FedMRL (49.7) and far less than FedKD. This confirms that FedEMoE delivers superior performance while incurring the smallest extra computational overhead.
> > >
> > > **Table I:** Computational cost to reach 35% accuracy on CIFAR-100 with 50 clients.
> > >
> > > |                                              | **Ours** | **FedMRL** | **FedKD**    | **FedProto** | **FedSA**    |
> > > |----------------------------------------------|----------|------------|--------------|--------------|--------------|
> > > | **Rounds to 35% Accuracy**                            | 17       | 38         | >100         | >100         | >100         |
> > > | **Extra computation (GFLOPs)**               | 21.1     | 49.7       | >262.3       | >41.7        | >42.7        |

---

### Official Review · Reviewer_Tbta · 2026-03-13

**Soundness:** 2
**Presentation:** 3
**Significance:** 3
**Originality:** 3
**Overall Recommendation:** 5
**Confidence:** 5

**Summary:**

This paper introduces FedEMoE, an Elastic Mixture of Experts architecture designed for heterogeneous federated learning to decouple personalized and generalized knowledge. The framework utilizes a multi-scale feature extraction mechanism with personalized experts to capture local nuances, alongside an elastic shared expert that adaptively resizes via weight spectrum analysis to facilitate knowledge transfer across varying model architectures. Experimental results across multiple datasets and heterogeneity settings indicate that this approach improves individual model accuracy compared to existing state-of-the-art methods.

**Compliance With Llm Reviewing Policy:**

Affirmed.

**Final Justification:**

As my primary concerns have been addressed during the rebuttal phase, I would like to increase my overall rating.

**Key Questions For Authors:**

1. My primary concern lies in the additional computational costs introduced by the Mixture of Experts (MoE) architecture and the specialized local update algorithms. Compared to existing personalized FL methods that rely on simpler parameter isolation (e.g., FedPAC [1]), the MoE structure may impose a significantly higher burden on local clients.

**Limitations:**

No. A more thorough analysis of the potential failure modes and limitations of FedEMoE is necessary to provide a balanced perspective on its practical deployment.

**Strengths And Weaknesses:**

**Strengths:**

**1. Organization and Clarity:** The manuscript is well-structured and follows a logical progression, ensuring the technical contributions are clear and easy to follow.

**2. Innovative Methodology:** The proposed FedEMoE framework offers a novel perspective on disentangling the model parameter space. It effectively addresses the dual challenge of balancing personalization and generalization within the heterogeneous federated learning paradigm.

**3. Comprehensive Evaluation:** The authors provide a solid theoretical foundation and extensive empirical evaluations across multiple benchmarks, convincingly demonstrating the performance gains of the proposed method over current state-of-the-art (SOTA) baselines.

**Weaknesses:**

**1. Analysis of Computational Overhead:** My primary concern lies in the additional computational costs introduced by the Mixture of Experts (MoE) architecture and the specialized local update algorithms. Compared to existing personalized FL methods that rely on simpler parameter isolation (e.g., FedPAC [1]), the MoE structure may impose a significantly higher burden on local clients.

**2. Practical Deployment Constraints:** Given that limited local computational resources are a primary bottleneck in federated learning, the absence of a detailed discussion and empirical evaluation regarding these costs hinders the assessment of the framework's practical viability. The manuscript would be significantly strengthened by an analysis of the trade-off between the performance gains of FedEMoE and its resource consumption (e.g., FLOPs, memory usage, and training time).

**Final Note:** If the authors can provide evidence or analysis to address these concerns during the rebuttal phase, I would be glad to increase my overall rating.

[1] Personalized Federated Learning with Feature Alignment and Classifier Collaboration, ICLR 2023.

---

> ### Author Rebuttal · Authors · 2026-03-30
>
> We thank the reviewer for valuable comments. In the following, we provide detailed explanations addressing each concern, and we believe the issues raised have been clarified.
>
> **R1:** Under the same experimental setting, we compare w & w/o MoE architecture on the computational costs, which **just increase a little** additional computational costs introduced by the MoE architecture.
> During training (both forward and backward propagation), the model without MoE = 0.961 GFLOPs, whereas the model with MoE averages ≈ 1.34 GFLOPs, attributable to the lightweight components of the MoE architecture.
> In a nutshell, **the computational overhead is acceptable by the corresponding gains in accuracy.**
>
>
> **R2:** As demonstrated in the newly added Table I, FedEMoE's local extra GFLOPs is 1.24 and its Total Time (93s) is faster than most baselines, including FedProto (105s) and FedTGP (145s). This high wall-clock efficiency is achieved because our elastic MoE utilizes sparse activation and parallel processing.
>
> Regarding memory consumption, our client-side heterogeneous models are constrained to closely match a standard ResNet-18. Because only a subset of sub-experts participates in the gradient calculation for any given sample, the peak local GPU memory usage remains well within the strict limits of standard edge devices.
>
> We also clarify that all resource-intensive operations (Weight Spectrum Analysis) are strictly offloaded to the server-side. The local clients only perform standard model updates and simple logit-level bidirectional knowledge distillation, keeping edge computation lightweight.
>
> When weighing this manageable resource consumption against the performance gains, FedEMoE offers an exceptional trade-off. For instance, while FedEMoE's total time (93s) is comparable to FedMRL (91s), FedEMoE achieves a striking 49.03% accuracy on CIFAR-100 with 20 clients, vastly outperforming FedMRL (39.19%) and FedProto (29.12%). We think this massive leap in personalization completely justifies the highly controlled resource overhead. We have included this table and detailed analysis in the revised manuscript.
>
> Table I: Comparison of computational overhead of different methods on CIFAR-100 with 20 clients.
>
> |                  | Total Time(s) | Local extra GFLOPs |
> |------------------|---------------|--------------------|
> | **FedKD**        | 352           | 2.61               |
> | **FedProto**     | 105           | 0.417              |
> | **FedMRL**       | 91            | 1.31               |
> | **FedTGP**       | 145           | 0.424              |
> | **FedSA**        | 112           | 0.426              |
> | **Ours**         | 93            | 1.24               |

---

> > ### Author Rebuttal · Reviewer_Tbta · 2026-04-04
> >
> > I thank the authors for their detailed clarifications and the additional experimental results, which support the practical deployment of the proposed method. As my primary concerns have been addressed during the rebuttal phase, I am going to increase my overall rating. I strongly recommend that the authors incorporate these new clarifications and empirical results into the future version of this manuscript.

---

> > > ### Author Response · Authors · 2026-04-05
> > >
> > > Thank you sincerely for recognition and raising your score. We have carefully revised the manuscript based on your comments and suggestions.

---

### Decision · Program_Chairs · 2026-04-30

**Decision:**

Accept (regular)

**Comment:**

This paper proposes an Elastic Mixture of Experts architecture designed for heterogeneous federated learning. The manuscript is well-structured and follows a logical progression, ensuring the technical contributions are clear and easy to follow. Besides, the theoretical analysis is sound and provides a novel perspective for analyzing convergence of MOE-based models. However, there have been may similar mechanisms of decoupling the konwledge into personal and global  part. To this end, I recommend this paper to accept.